# Comprehensive evaluation of deconvolution methods for human brain gene expression

Gavin J. Sutton[1], Daniel Poppe[2,3], Rebecca K. Simmons[2,3], Kieran Walsh[1], Urwah Nawaz[4], Ryan Lister [2,3], Johann A. Gagnon-Bartsch[5] & Irina Voineagu [1,6 ✉]

Transcriptome deconvolution aims to estimate the cellular composition of an RNA sample from its gene expression data, which in turn can be used to correct for composition differences across samples. The human brain is unique in its transcriptomic diversity, and comprises a complex mixture of cell-types, including transcriptionally similar subtypes of neurons. Here, we carry out a comprehensive evaluation of deconvolution methods for human brain transcriptome data, and assess the tissue-specificity of our key observations by comparison with human pancreas and heart. We evaluate eight transcriptome deconvolution approaches and nine cell-type signatures, testing the accuracy of deconvolution using in silico mixtures of single-cell RNA-seq data, RNA mixtures, as well as nearly 2000 human brain samples. Our results identify the main factors that drive deconvolution accuracy for brain data, and highlight the importance of biological factors influencing cell-type signatures, such as brain region and in vitro cell culturing.

[1] School of Biotechnology and Biomolecular Sciences, University of New South Wales, Sydney, NSW, Australia. [2] Harry Perkins Institute of Medical Research, QEII Medical Centre and Centre for Medical Research, The University of Western Australia, Perth, WA, Australia. [3] Australian Research Council Centre of Excellence in Plant Energy Biology, School of Molecular Sciences, The University of Western Australia, Perth, WA, Australia. [4] Adelaide Medical School, Robinson Research Institute, University of Adelaide, Adelaide, SA, Australia. [5] Department of Statistics, University of Michigan, 1085 South University Ave, Ann Arbor, MI 48109, USA. [6] Cellular Genomics Futures Institute, University of New South Wales, Sydney, NSW, Australia. ✉email: i.voineagu@unsw.edu.au

Human tissues are mosaics of cell-types and subtypes, diverse in their functionalities and express distinct sets of genes. Consequently, gene expression measurements in any tissue sample result from two main factors: gene expression levels within constituent cell-types, and the relative abundance of these cell-types in the sample[1,2]. The relative abundance of cell-types (i.e. cellular composition) in turn depends on both biological[3–6] and technical factors[7].

To circumvent the confounding effect of cellular composition, gene expression measurements could in principle be carried out by experimentally isolating individual cell-types by laser capture micro-dissection[8,9], cell sorting[10–12], or single-cell and single-nucleus RNA-seq (scRNA-seq and snRNA-seq, respectively)[13]. In practice, these approaches are limited in feasibility and cost effectiveness for human brain transcriptome studies that require large sample sizes (hundreds to thousands of samples), such as eQTL studies or gene expression studies aiming to identify low-magnitude changes in disease samples.

Several methods for in silico deconvolution have been developed, which can estimate the cellular composition of a tissue sample from its gene expression profile (reviewed in Avila Cobos et al.[1]). In silico deconvolution offers the opportunity to leverage scRNA-seq data to obtain deeper insights into bulk tissue transcriptomes generated through large-scale studies, such as GTEx[14], PsychEncode[15], the Common Mind Consortium[16], and BrainSpan[17].

Deconvolution methods fall into two main categories: partial deconvolution (including enrichment approaches), and complete deconvolution (see the "Methods" section), and are conceptually similar for any tissue and any type of molecular data (transcriptome, methylome, proteome, etc.). However, the complexity of cellular composition and the transcriptome similarity across cell-types varies widely across tissues. Most deconvolution methods have been developed for, or assessed on, blood/immune and tumour samples[18–20], with limited assessment of their performance across tissues[2]. Recent benchmarking studies have assessed the role of technical and biological factors in transcriptome deconvolution[21,22], but how these observations hold across tissues remains unclear. For example, the human brain expresses the highest diversity of alternative splicing isoforms and non-coding RNAs[23], with single-cell sequencing now discovering hundreds of cell-types and cell-subtypes[24]. We thus begin to address the question of tissue-specific properties of transcriptome deconvolution by focusing on the human brain. The main biological factors that influence the cellular composition of brain samples (e.g. region, developmental stage, age[4,5]), and the technical factors involved (e.g. dissection protocol[7]) are distinct from those influencing cellular composition in blood. Furthermore, pure populations of cells from adult human brain are challenging to obtain, unlike blood or tumour cells. As a result, cell-type signature data are often obtained from a different brain region[25], species[26,27], and/or a different developmental stage[7] than the bulk brain samples. Alternatively, cells cultured in vitro have been used[19]. Whether such choices influence the accuracy of brain cell-type composition estimates is unknown. In addition, gene expression changes in most psychiatric disorders, similarly to effect-sizes of common variants, are of low magnitude[28]. Therefore, to serve as useful covariates, cell-type composition estimates need to discriminate small differences in cellular composition[3]. While a few studies have proposed methods focussed on brain tissue[5,7,27,29–31], a comprehensive comparative assessment of the performance of deconvolution methods on brain transcriptome data is currently lacking.

Here, we perform a comprehensive evaluation of brain transcriptome deconvolution by assessing the performance of eight algorithms (four partial deconvolution, two enrichment, and two complete deconvolution methods). For the partial deconvolution methods, we evaluate the effects of combining them with nine brain cell-type signature datasets that differ in biological properties (cultured cells, immuno-purified cells, or cross-species) or technical factors affecting RNA sequencing (snRNA-seq, scRNA-seq, bulk RNA-seq, or CAGE-seq). We benchmark deconvolution accuracy using a diverse set of mixtures, including in silico mixtures of single-cell and single-nucleus transcriptomes, pure immuno-panned cell-types, mixtures of RNA extracted from pure populations of neurons and glial cells, as well as large-scale brain transcriptome data from the GTEx[14] and PsychENCODE[15,32] consortia. Our results show that partial deconvolution methods, particularly CIBERSORT, outperform complete deconvolution methods on human brain data. For partial deconvolution, cell-type signature data is the most important parameter, with the main factors influencing performance being biological (brain region and in vitro cell culturing) rather than technical. We also assess methods for correcting cell-type composition differences in differential expression analyses, and determine the magnitude of cell-type composition differences that can be effectively corrected for. Finally, we deconvolve large-scale gene expression data from the GTEx and PsychENCODE consortia, and highlight the importance of assessing deconvolution accuracy on each brain dataset; for this purpose, we provide a user-friendly web tool that implements the best performing methods identified in our benchmark (https://voineagulab.shinyapps.io/BrainDeconvShiny/).

## Results
To benchmark deconvolution methods for brain transcriptome data, we selected widely employed methods, and where possible included methods developed for brain data (Table 1). For partial deconvolution, we selected: CIBERSORT (CIB), a highly cited deconvolution method initially optimised for immune cell-types[18]; DeconRNASeq (DRS)[33], which implements the non-negative least-squares approach employed by the PsychENCODE consortium[15]; MuSiC (MUS)[34], which is a single-cell-based deconvolution approach accounting for individual- and cell-specific expression variability in the signature; and dtangle (DTA)[35]. For enrichment-based methods, we selected xCell[19], which has been recently applied by the GTEx consortium[36], and BrainInABlender[7], which was specifically developed for brain-derived data. Among complete deconvolution methods, we included Linseed[37], which extends previous methods[38,39], and the co-expression-based approach developed for brain data by Kelley et al.[5] (which we term "Coex" for short).

**Assessment of deconvolution accuracy across methods**. To assess deconvolution accuracy, we simulated data with known cell-type proportions using three adult human brain datasets: two snRNA-seq datasets, Velmeshev et al.[40] (VL: 24,646 nuclei, 10X Chromium, Fig. 1) and Hodge et al. from the Human Cell Atlas[41] (CA: 11,314 nuclei, Smart-seq2, Supplementary Fig. 1); as well as scRNA-seq data from Darmanis et al.[13] (DM: 297 cells, Smart-seq, Supplementary Fig. 2). For each dataset, 100 mixtures were simulated as the average expression of 500 randomly sampled nuclei (VL, CA; see the "Methods" section) or 100 randomly sampled cells (DM; see the "Methods" section). Corresponding cell-type signatures were generated as the average expression within each cell-type (see the "Methods" section).

We first estimated cell-type proportions in these mixtures using CIBERSORT, DeconRNASeq, dtangle, and MuSiC, and enrichment scores using xCell and BrainInABlender, evaluating six major brain cell-types: neurons, astrocytes, oligodendrocytes, oligodendrocyte precursor cells (OPCs), microglia, and endothelia. Focusing on mixtures generated from the largest dataset (VL), we found that deconvolution accuracy was very high for

**Table 1 Description of algorithms benchmarked in this study.**

| Algorithm | Class | Signature | Foundation | Output | Citation |
|---|---|---|---|---|---|
| DeconRNASeq | Deconvolution | User-specified | Non-negative least squares | Proportions | Gong et al.[33] |
| CIBERSORT | Deconvolution | User-specified | Support vector regression | Proportions | Newman et al.[18] |
| Dtangle | Deconvolution | User-specified | Linear mixing model | Proportions | Hunt et al. (2019) |
| MuSiC | Deconvolution | User-specified (single-cell only) | Weighted non-negative least squares | Proportions | Wang et al. (2019) |
| Linseed | Deconvolution | None | Simplex topology | Proportions of unlabelled cell-types | Zaitsev et al.[37] |
| BrainInABlender | Enrichment | In-built (human and mouse brain) | Average scaled expression of marker genes | Enrichment | Hagenauer et al.[7] |
| xCell | Enrichment | In-built (cultured human brain cells) | Gene set enrichment analysis | Enrichment | Aran et al.[19] |
| Coex | Enrichment | None | Weighted gene co-expression network analysis | Enrichment for unlabelled cell-types | Kelley et al.[5] |

CIBERSORT (mean $r$ across cell-types = 0.87), MuSiC (0.82), and dtangle (0.87), but lower for DeconRNASeq (0.50) (Fig. 1B, left, and Supplementary Figs. 3, 4). For the two enrichment algorithms, BrainInABlender's accuracy was moderately high but inconsistent across cell-types, while xCell poorly estimated cell-type abundance ($r = -0.06$ and 0.02 for neurons and astrocytes, respectively); Fig. 1C, and Supplementary Fig. 4. These observations were replicated in both the CA-based (Supplementary Figs. 1, 5, 6) and DM-based simulations (Supplementary Fig. 2), suggesting that (a) deconvolution of major brain cell-types is accurate across a range of partial deconvolution algorithms, with CIBERSORT generally performing best based on both $r$ and normalised mean absolute error (nmae) values (Fig. 1, Supplementary Figs. 1–6) and (b) enrichment methods are less accurate than partial deconvolution methods, with xCell showing particularly low accuracy.

We next assessed deconvolution accuracy on five in vitro RNA mixture samples of known composition (Supplementary Fig. 7) and 21 RNA samples from pure populations of cells immuno-panned with cell-type-specific antibodies[42] (Supplementary Fig. 8). In both cases, the deconvolution accuracy was very high when the signature was derived from the same dataset as the mixtures. For RNA mixtures, the normalised mean absolute error was 0.035, 0.043, and 0.11 for CIBERSORT, DeconRNASeq, and dtangle, respectively (Supplementary Fig. 7). For RNA extracted from sorted cells, the immuno-panned cell-type was identified on average as 96.3%, 93.0%, and 92.6% abundant by CIBERSORT, DeconRNASeq, and dtangle, respectively (Supplementary Fig. 8).

Next, we explored how including cell sub-types affected deconvolution accuracy. First, we used broad neuronal sub-types, i.e. excitatory and inhibitory neurons (Fig. 1B middle, Supplementary Figs. 3, 9), and found that deconvolution accuracy was high ($r > 0.8$ for all algorithms), with CIBERSORT performing best ($r = 0.94$ and 0.95 for excitatory and inhibitory, respectively). The accuracy for the other cell-types was largely unaffected by neurons being sub-classified (Fig. 1B, middle, Supplementary Figs. 3, 9). This result was replicated in the CA-based simulations (Supplementary Figs. 1, 5, 10).

When including all cell sub-types detected in the VL dataset (11 neuronal and 2 astrocyte sub-types), deconvolution with CIBERSORT remained accurate ($r > 0.8$) for most cell populations (Fig. 1B, right, Supplementary Figs. 3, 11, 12). However, we noted that the cell-types with comparatively lower estimation accuracy had two properties: low abundance in the mixture (<2%) and high collinearity (gene expression correlation with another cell-type $rho > 0.95$); Supplementary Figs. 13, 14. This observation was replicated in the CA dataset with most cell sub-types being accurately deconvolved ($r > 0.8$; Supplementary Figs. 1, 5, 15, 16) and collinearity being the main factor that led to reduced accuracy (Supplementary Figs. 17, 18).

We finally explored how deconvolution was affected when the cell-type signature matrix was incomplete. To do so, we deconvolved VL-derived mixtures when omitting one cell-type at a time from the signature. We found that when an abundant cell-type was missing (Neurons, 87.4% mean abundance), the deconvolution accuracy was substantially reduced (mean $r$ was reduced from 0.85 to 0.41, and normalised mean absolute error increased from 0.33 to 10.3). However, when lowly abundant cell-types were missing from the signature, the effect on deconvolution was minimal (Supplementary Fig. 19). We then tested the effect of removing a sub-type of neurons, excitatory or inhibitory neurons, which are highly correlated in expression ($rho = 0.92$). Deconvolution accuracy was reduced to a lesser extent than when all neurons were missing: $r$ was reduced from 0.87 to 0.71 when excitatory neurons were missing, and from 0.86 to 0.76 when inhibitory neurons were missing (Supplementary Fig. 19).

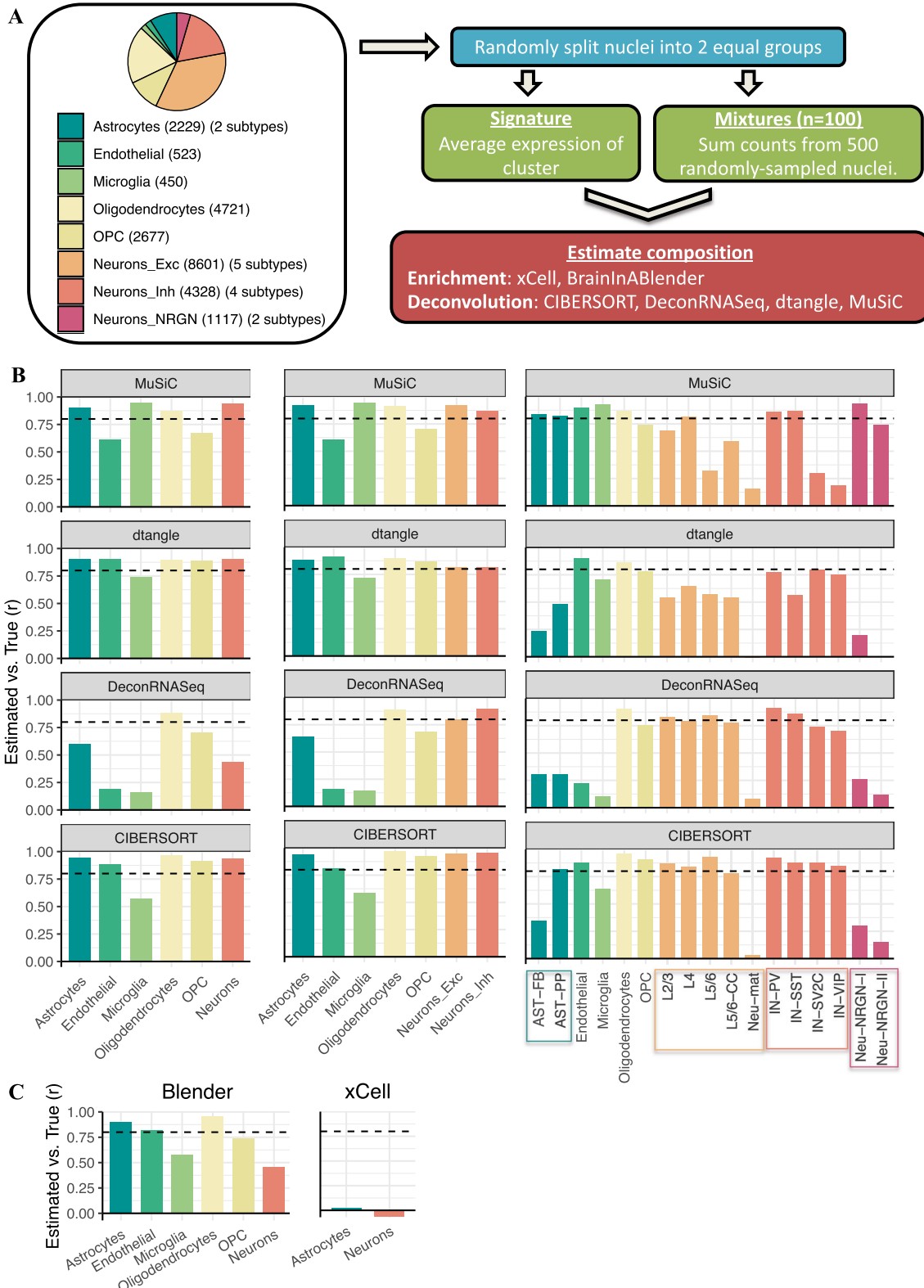

**Fig. 1 Deconvolution accuracy across methods. A** Simulation design. Single-nucleus RNA sequencing data was acquired from Velmeshev et al. and used to create 100 in silico mixtures with known proportions (see the "Methods" section). *Left:* Piechart displaying the composition of the dataset, where *n* is the number of cells per cell-type. The number of sub-types is listed in between brackets. *Right:* analysis outline. OPC oligodendrocyte precursor cells. Neurons_Exc, Neurons_Inh, and Neurons_NRGN: excitatory, inhibitory, and NRGN[+] neurons, respectively. DRS DeconRNASeq, CIB CIBERSORT, Blender BrainInABlender. **B** Barplots of Pearson correlation coefficients (*r*) between true and estimated cell-type proportions in 100 in silico mixtures. *Left:* cells are grouped by major cell-types; *middle:* excitatory and inhibitory neuron subtypes are included in the signature; *right:* all cell-subtype labels are used in the signature. **C** Barplots of Pearson correlation coefficients (*r*) between true proportion and cell-type enrichment scores. Dotted horizontal lines: $r = 0.8$.

**The biological properties of cell-type signature data strongly influence deconvolution**. We hypothesised that the brain cell-type signature data could have a major impact on the deconvolution outcome. This has been previously reported for deconvolution of blood transcriptomes[43], and is supported by our observation that xCell performed significantly worse than the other deconvolution methods (noting that its signature data is built-in).

To investigate how the properties of the signature data influence the deconvolution outcome, we deconvolved the human brain snRNA-seq mixtures (VL) using cell-type signature data from several datasets (see the "Methods" section; Fig. 2) with different sequencing methods and various sources of brain tissue: human brain snRNA-seq (CA[41], NG[44], LK[45]); scRNA-seq from the human (DM[13]) or mouse (TS[46]) brain; bulk RNA-seq of immuno-panned cells from the human (IP[42]) or mouse brain (MM[47]); or CAGE-seq from cultured human brain cells (F5[48]).

We found that the choice of cell-type signature data strongly affected the deconvolution accuracy. Using data from cultured brain cells (F5) dramatically reduced the accuracy (Fig. 2A, B); this likely explains the poor performance of xCell, which has F5 as the main built-in signature. Using signature data from the mouse brain (TS, MM) also reduced the deconvolution accuracy (Fig. 2A, B). These observations were consistent across deconvolution algorithms (Supplementary Fig. 20) and were replicated when deconvolving in silico mixtures based on the CA and DM data (Supplementary Figs. 21, 22), as well as with deconvolution of broad neuronal subtypes (Supplementary Figs. 23, 24). Conversely, when deconvolving RNA mixtures of known composition from cultured cells (see the "Methods" section), using the cultured-cell F5 signature data performed the best despite the difference in sequencing technology (CAGE-seq vs. RNA-seq; Supplementary Fig. 7).

Overall, these data demonstrate that the biological properties of the cell-type signature data strongly impact the deconvolution accuracy, having a more pronounced effect than the sequencing methods, and highlight in vitro culturing of brain cells as an important biological factor.

**The effect of compartment-specific genes on deconvolution accuracy**. Since most single-cell data from the adult human brain are generated using single-nucleus RNA-seq, while bulk RNA-seq is based on total RNA, we next investigated whether compartment-specific genes (i.e. those either enriched or depleted from the nucleus) influence the outcome of deconvolution. For this purpose, we generated paired bulk RNA-seq and nuclear RNA-seq from five frozen brain tissue samples (see the "Methods" section), as well as snRNA-seq from the same brain samples. We identified 2808 compartment-specific genes as those differentially expressed between the nuclear and total bulk RNA-seq (FDR < 0.05, |FC| > 1.3; Supplementary Data 1). We then carried out several deconvolution analyses with and without filtering-out the compartment-specific genes.

Firstly, we deconvolved the 21 bulk RNA-seq samples from sorted brain cells[42], where true cell-type composition is known (i.e., each sample is expected to be a nearly pure cell-type, with some experimental variability of the sorting efficiency). We deconvolved these data with either the matched cell-type signature (derived from this sorted dataset; IP), an scRNA-seq signature (DM), and four snRNA-seq signatures (VL, CA, NG, and LK); Supplementary Data 2. When using the IP signature, the immuno-panned cell-type was estimated as >80% abundant in all samples. Thus, we assessed the proportion of samples in which the sorted cell-type was correctly identified (i.e., estimated proportion >80%) using the scRNA-seq and snRNA-seq

signatures, with or without filtering out compartment-specific genes. We found that the snRNA-seq signatures performed well even prior to filtering out compartment-specific genes, correctly identifying the sorted cell-type in an average of 86% of samples (71–95%). As expected, the single-cell-based signature (Supplementary Fig. 25) performed somewhat better, identifying the sorted cell-type in an average of 90% of samples. Removing compartment-specific genes further improved the outcome for snRNA-seq signatures: the sorted cell-type was correctly identified in an average of 88% of samples (86–95%); Supplementary Fig. 25, eliminating the difference between the scRNA-seq and snRNA-seq signatures.

To increase the complexity of the deconvolution task, we asked how accurately the five whole-tissue samples were deconvolved when using snRNA-seq data from the same individuals, as compared to a whole-cell-based signature (Supplementary Fig. 25). In this case, if compartment-specific genes were not removed from the cell-type signature, the correlation between cell-type proportions estimated using the snRNA-seq signature and the whole-cell signature was modest ($r = 0.27$). However, the correlation improved substantially by filtering out compartment-specific genes ($r = 0.98$) suggesting that this filtering approach should be considered when using snRNA-seq-based cell-type signatures.

**Reference-free complete deconvolution methods are less effective on brain gene expression data than partial deconvolution methods**. Since we observed a strong effect of the choice of reference signature data on the brain deconvolution outcome, and recent studies have proposed reference-free approaches to cell-type composition estimation[37,38,49], we assessed the performance of two such methods on brain data. Linseed, a complete deconvolution algorithm[37], proposes to identify cell-type-specific genes by representing the expression vector of each gene as a point in N-dimensional space (where N is the number of samples). It also proposes using singular value decomposition (SVD) to determine the number of cell-types from the mixture data. An alternative approach, Coex[50], employs co-expression networks to identify modules of co-expressed genes enriched for specific cell-type markers, and then uses the module eigengene values as cell-type-enrichment scores[5].

When applying Linseed to in silico mixtures generated by random sampling from the three benchmarking datasets (VL, CA, DM), we found that the SVD approach did not correctly identify the number of cell-types in the mixture (see the "Methods" section; Supplementary Fig. 26). With the correct number of cell-types specified, Linseed performed less accurately than the partial deconvolution methods, with $r > 0.8$ achieved for only two cell-types for VL and CA, and none of the cell-types for DM (Fig. 3B, Supplementary Fig. 27). On the RNA mixtures however, Linseed performed very accurately ($r = 1$; Supplementary Fig. 28). Since Linseed relies on the detection of genes represented by points with "extreme" positions in the $k-1$ dimensional simplex, we hypothesised that the difference in its performance between the datasets likely results from the wider distribution of cell-type proportions in the RNA mixtures (neuronal proportions: 0–100%), than in the mixtures generated by random sampling from real brain datasets. To test this hypothesis, we generated mixtures with a broad range of cell-type abundances using VL and CA (see the "Methods" section; Fig. 3D, top and bottom). The performance of Linseed improved markedly on both datasets using these controlled in silico mixtures (Fig. 3E top and bottom, Supplementary Figs. 29, 30), with the SVD approach also better identifying the number of cell-types in the mixture (Supplementary Fig. 26).

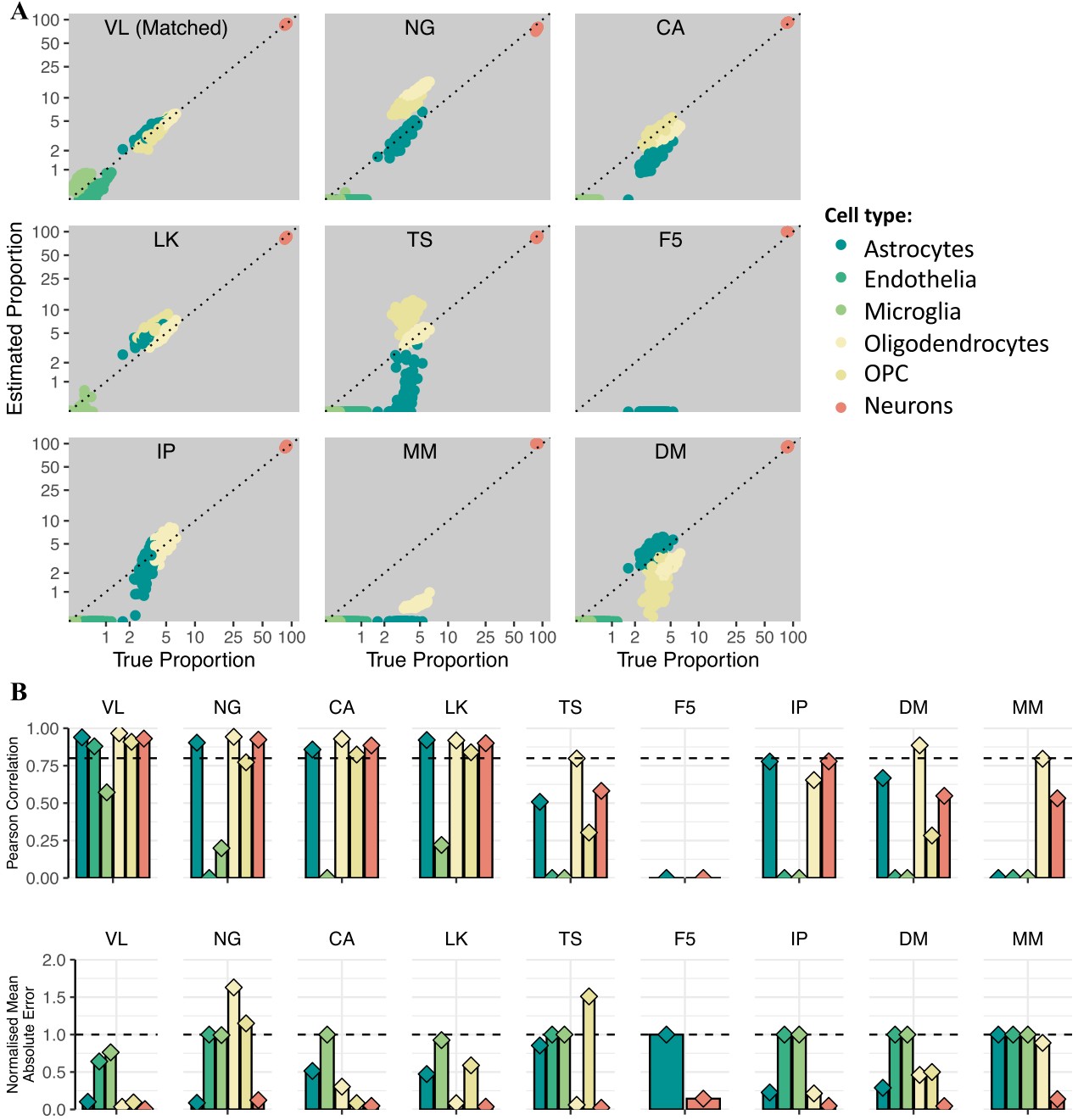

**Fig. 2 Effect of signature choice on deconvolution accuracy. A** Scatterplots of true and CIBERSORT-estimated proportions in VL-derived in silico mixtures, for nine different signatures. *Matched:* the signature and mixture were derived from the same dataset. Plot titles represent the signature used in deconvolution, as follows: VL Velmeshev, NG Nagy, CA Human Cell Atlas, LK Lake, TS Tasic, F5 FANTOM5, IP immuno-paned, MM mouse immuno-panned, DM darmanis. See the "Methods" section for further details about signatures. Dotted line: $y = x$. **B** Barplots of Pearson correlation ($r$) for all cell-types and signatures presented in (**A**). Dotted line: $r = 0.8$. **C** Barplots of normalised mean absolute error (*nmae*) for all cell-types and signatures presented in (**A**). OPC oligodendrocyte precursor cells. *Dotted line:* nmae = 1.

We found that Coex also performed significantly less accurately than the partial deconvolution methods on the randomly sampled mixtures (Fig. 3C, Supplementary Fig. 27). Since the co-expression network approach also relies on gene expression co-variation driven by differences in cell-type proportions, its performance improved on simulations with a wider range of cell-type proportions, but did not achieve accurate deconvolution for all cell-types (Fig. 3F, top and bottom, Supplementary Figs. 29, 30).

These data suggest that complete deconvolution methods are less effective than partial deconvolution methods for brain cell-types, particularly since the performance of these algorithms is related to the variance in cellular composition of the dataset, which is not known a priori.

**Assessment of the interplay between cell-type composition and differential gene (DE) expression analyses.** We next investigated

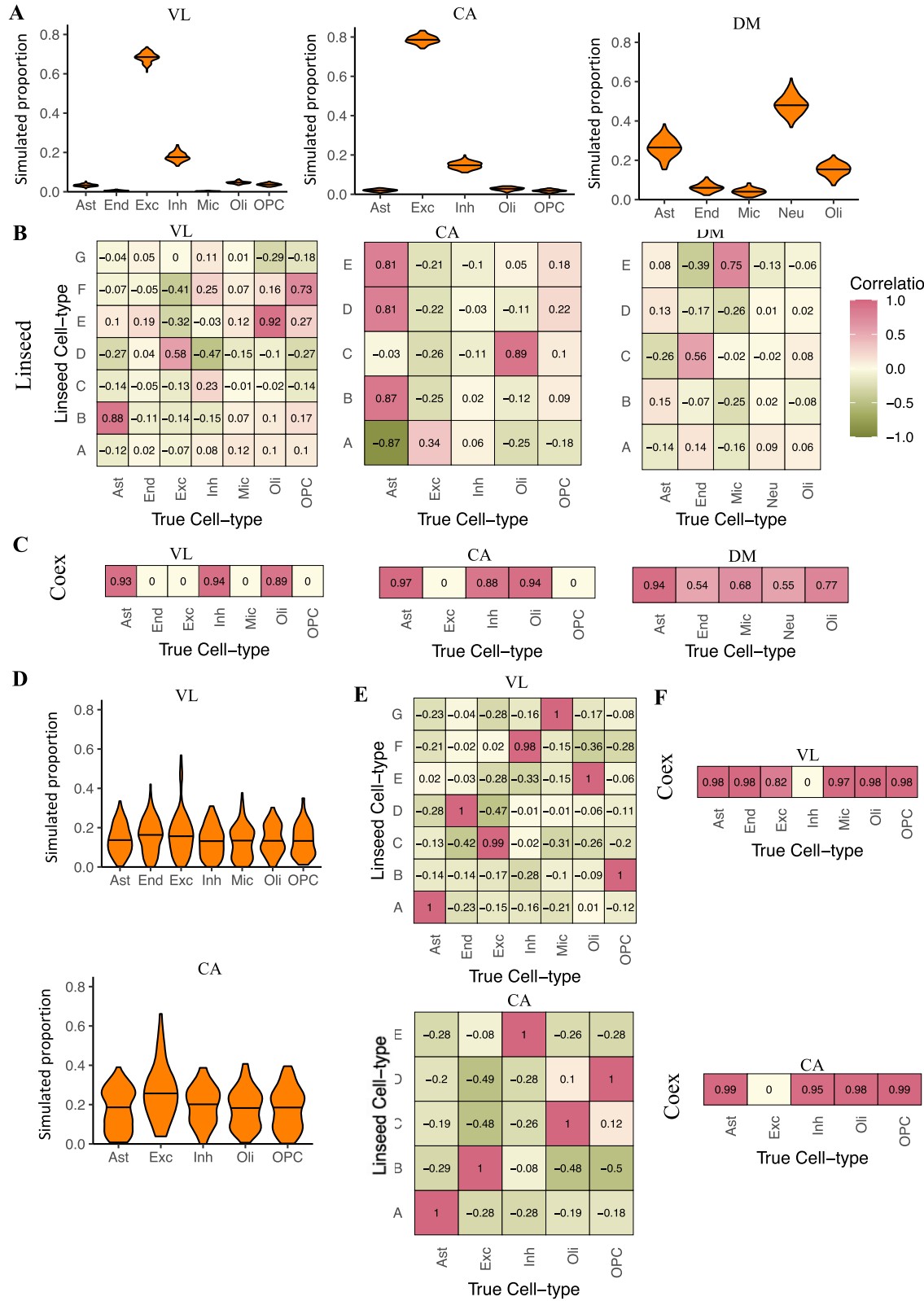

how cellular composition influences DE analyses, in particular: (i) how much should cell-type composition differ between two groups of brain samples to lead to false positive results in DE analyses, and (ii) what is the best approach for correcting cell-type composition differences in DE analyses?

We used the CA dataset[41] (Smart-seq2, high coverage per gene) to generate simulated data for two-group DE analyses. Each dataset contained two groups of 50 samples (group A and group B). The proportion of one of the cell-types (excitatory neurons) was simulated as either higher or lower in group B than in group A by 0–40% (see the "Methods" section; Fig. 4). We then carried out DE comparing group B to group A, using either a linear model (LM) or a generalised LM as implemented in DESeq2[51] with and without correction for cellular composition. False-

**Fig. 3 Reference-free deconvolution. A** Violin plots of the distribution of true cell-type proportions in in silico simulations derived from data from Velmeshev et al. (VL), the Human Cell Atlas (CA), and Darmanis et al. (DM) (left, middle, and right, respectively). The width of the violin indicates point density. *Black horizontal bar:* median. Ast astrocytes, End endothelia, Exc excitatory neurons, Inh inhibitory neurons, Neu neurons, Oli oligodendrocytes, OPC oligodendrocyte precursors. **B** and **C** Heatmaps of Pearson correlation coefficients between estimated and true cell-type proportions for random simulations based on VL, CA, and DM. **B** Linseed; *y*-axis: Cell-types defined by Linseed; *x*-axis: true cell-type in simulated data. **C** Coex; for each cell-type the true vs. estimated correlation coefficient is displayed for the co-expression module assigned to that cell-type based on marker enrichment (see the "Methods" section); zero values represent cases where no co-expression module was assigned to the corresponding cell-type. **D** Violin plots of the distribution of cell-type abundances in simulations with wide cell-type ranges based on VL (top) and CA (bottom). **E** and **F** Heatmaps of Pearson correlation coefficients between estimated and true cell-type proportions for simulations with wide cell-type ranges based on VL and CA. **E** Linseed; *y*-axis: Cell-types defined by Linseed; *x*-axis: true cell-type in simulated data. **F** Coex; for each cell-type the true vs. estimated correlation coefficient is displayed for the co-expression module assigned to that cell-type based on marker enrichment (see the "Methods" section); zero values represent cases where no co-expression module was assigned to the corresponding cell-type.

positives driven by cellular composition were defined as genes differentially expressed at a false discovery rate (FDR) < 0.05 (see the "Methods" section).

We found that without correction, differences in cellular composition of <5% between the sample groups led to fewer than 10 false-positive DE genes. However, above 5% the number of false-positive genes increased steeply with the difference in cellular composition, reaching >10,000 at a 20% difference in cellular composition (Fig. 4A). Inclusion of excitatory neuron proportions as a covariate in the LM effectively eliminated false-positive genes (Fig. 4A). We did not observe any additional benefit when using a spline matrix as covariate, while quadratic regression was less effective than linear regression at larger composition confounds (Fig. 4A). We also found that including cellular composition estimates in DESeq2 was similarly effective at eliminating false-positives (Fig. 4A). As expected, markers of excitatory neurons were enriched among downregulated genes when the proportion of this cell-type was reduced in the test group, but enriched among upregulated genes when the proportion was increased (Fig. 4B).

We next investigated the more challenging case where, in addition to differences in cell-type composition, there are true differences in gene expression between the two groups. To this end, we simulated data with differences in cell-type composition as above, while also introducing gene expression changes in a set of 200 genes, of which 100 are markers of excitatory neurons and 100 are non-marker genes (see the "Methods" section). Several sets of simulations were generated with a mean expression difference between groups of 1.1-, 1.3-, 1.5- or 2-fold.

To quantify how effectively cellular composition was corrected for, we calculated a discriminatory power metric: the fraction of the 200 perturbed genes that were in the top 200 most significant DE genes. This measure rewards true-positives while penalising false-positives. We found that without correction, the discriminatory power decreased with the magnitude of cell-type composition difference between the two groups (Fig. 4C; uncorrected). Correction for cell-type composition was effective at restoring discriminatory power for gene expression differences of 1.5-fold when composition differences were up to 12.5% (Fig. 4D; corrected). As expected, for expression differences of lower magnitude (1.1 and 1.3) the effective correction range was narrower (6.3% and 6.9%, respectively), while for stronger expression differences (2-fold) the effective correction range was wider (25.7%); Fig. 4D, Supplementary Fig. 31. All correction approaches performed similarly in this analysis, with the exception of spline regression which we found to be less effective (Fig. 4C, D, Supplementary Fig. 31).

We also investigated whether the cell-type where differential expression occurs can be uncovered through deconvolution. To this end, we used CIBERSORTx[52], which takes a bulk mixture and estimates gene expression values in each cell-type present in the signature data; these cell-type-specific expression values can

then be used to carry out cell-type-specific DE analyses. We thus simulated data with 1.5-fold change in expression specific to a given cell-type, with or without superimposed differences in cell-type composition between the two groups (Fig. 5A), and tested whether genes were identified as DE in the correct cell-type. As above, the 1.5-fold expression difference was simulated for 200 genes, of which 100 are markers of the perturbed cell-type and 100 are non-marker genes.

The expression difference was first simulated in excitatory neurons (Fig. 5B, C). In the absence of confounding cell-type composition differences between the two groups, more than 95% of the perturbed excitatory marker genes were detected as DE in the correct cell-type (excitatory neurons), while for the non-marker genes ~45% were detected as DE in excitatory neurons and another ~30% were incorrectly detected as DE in inhibitory neurons (Fig. 5B). The false-positive rate (i.e. the fraction of non-perturbed genes detected as DE) was 0% (Fig. 5C). Similar results were observed when the gene expression change was modelled in inhibitory neurons (Fig. 5D, E).

When a composition difference was superimposed (~10% increase in excitatory neurons; see the "Methods" section), the true-positive rate was unchanged except for upregulated marker genes, where it was reduced, likely due to the fact that the composition change and the expression change were confounded (both variables were higher in group B vs. group A). The false-positive rate was <12% in all cell-types, thus drastically reduced relative to no correction for cell-type composition (32%), but higher than when correcting for composition differences in a standard linear model (0%) (Fig. 5B, C). Similar results were observed when the gene expression change was modelled in inhibitory neurons (Fig. 5D, E).

Finally, we assessed cell-type-specific DE in data from 15 autism spectrum disorder (ASD) and 11 control samples from Velmeshev et al.[40], comparing results from cell-type-specific pseudo-bulked data versus that estimated by CIBERSORTx (see the "Methods" section). Although no genes were significantly differentially expressed after multiple-testing correction, fold-changes determined in pseudo-bulk data correlated significantly with those estimated by CIBERSORTx for three of the four cell-types (see the "Methods" section; Supplementary Fig. 32).

Overall, these results suggest that using cell-type-specific gene expression for DE analyses is effective at detecting DE genes in the right cell-type when the gene expression and composition changes are not confounded, but this comes at the expense of a moderate increase in false-positives. Furthermore, low sample sizes and gene expression differences of low magnitude, as in the case of the pseudo-bulk ASD *vs.* control analysis, reduce the power of detecting cell-type-specific DE genes.

**Cell-type composition estimates in large-scale human brain transcriptome data.** We next evaluated the performance of brain

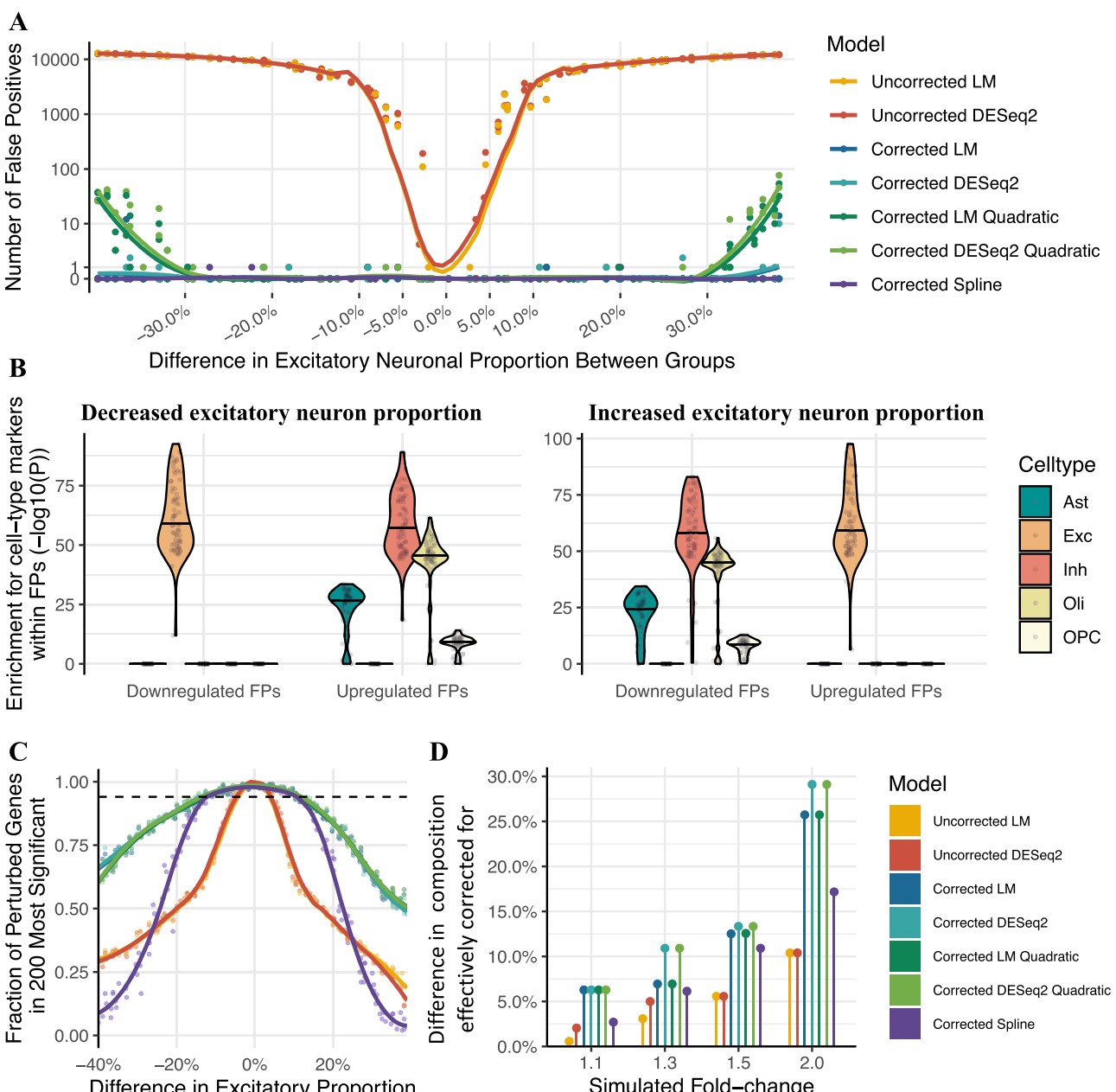

**Fig. 4 Effect of brain cell-type composition on Differential expression (DE) analyses. A** Scatterplot of the number of false positive genes versus the simulated difference in excitatory neuron proportion between two groups of 50 samples. Each point represents a different simulated dataset. DE was assessed with either a linear model (LM) or DESeq2. Models labelled as "corrected" adjusted excitatory neuronal proportion as a covariate. *Coloured lines:* local regression line. **B** Cell-type marker enrichment within false positive genes. Each point represents a single simulated dataset. The width of the violin indicates point density, and the horizontal black bar indicates the median. *y*-axis: enrichment *p*-value (one-sided Fisher test, no correction for multiple testing); Methods. FPs false positive genes, Ast Astrocytes, Exc excitatory neurons, Inh inhibitory neurons, Oli oligodendrocytes, OPC oligodendrocyte precursor cells. **C** Scatterplot of the discriminatory power, i.e. fraction of the 200 perturbed genes in the top 200 most significantly differentially expressed genes (*y*-axis) versus simulated difference in excitatory neuron proportion between sample groups (*x*-axis) for simulated 1.5-fold expression differences. *Coloured lines*: local regression line. *Dotted line:* expected discriminatory power, i.e. 0.95 times the discriminatory power in the absence of cell-type composition differences between groups. **D** Model robustness to cell-type composition differences across a range of fold-changes, quantified as the smallest composition change where discriminatory power fell below its expected value. This is defined as 0.95 times the discriminatory power of an uncorrected linear model in a simulation with no composition confound (calculated separately for each simulated fold-change).

gene expression deconvolution on large-scale datasets, focussing on a dataset of control individuals (GTEx[14], *n* = 1671 samples; see the "Methods" section), and a dataset of autism spectrum disorder (ASD) cases and controls (PsychENCODE; Parikshak et al.[32], *n* = 251 samples; see the "Methods" section). The GTEx data included samples from cerebellum (*n* = 309), cerebral cortex (*n* = 408), subcortical regions (*n* = 863) and spinal cord (*n* = 91);

the Parikshak et al. dataset included samples from cerebellum (*n* = 84) and cerebral cortex (*n* = 167).

We assessed all combinations of four partial deconvolution methods and nine cell-type signatures, the two enrichment methods, and coex as a complete deconvolution method (Supplementary Data 3). We also generated an additional signature (MultiBrain) by merging CA, IP, DM, NG, and VL

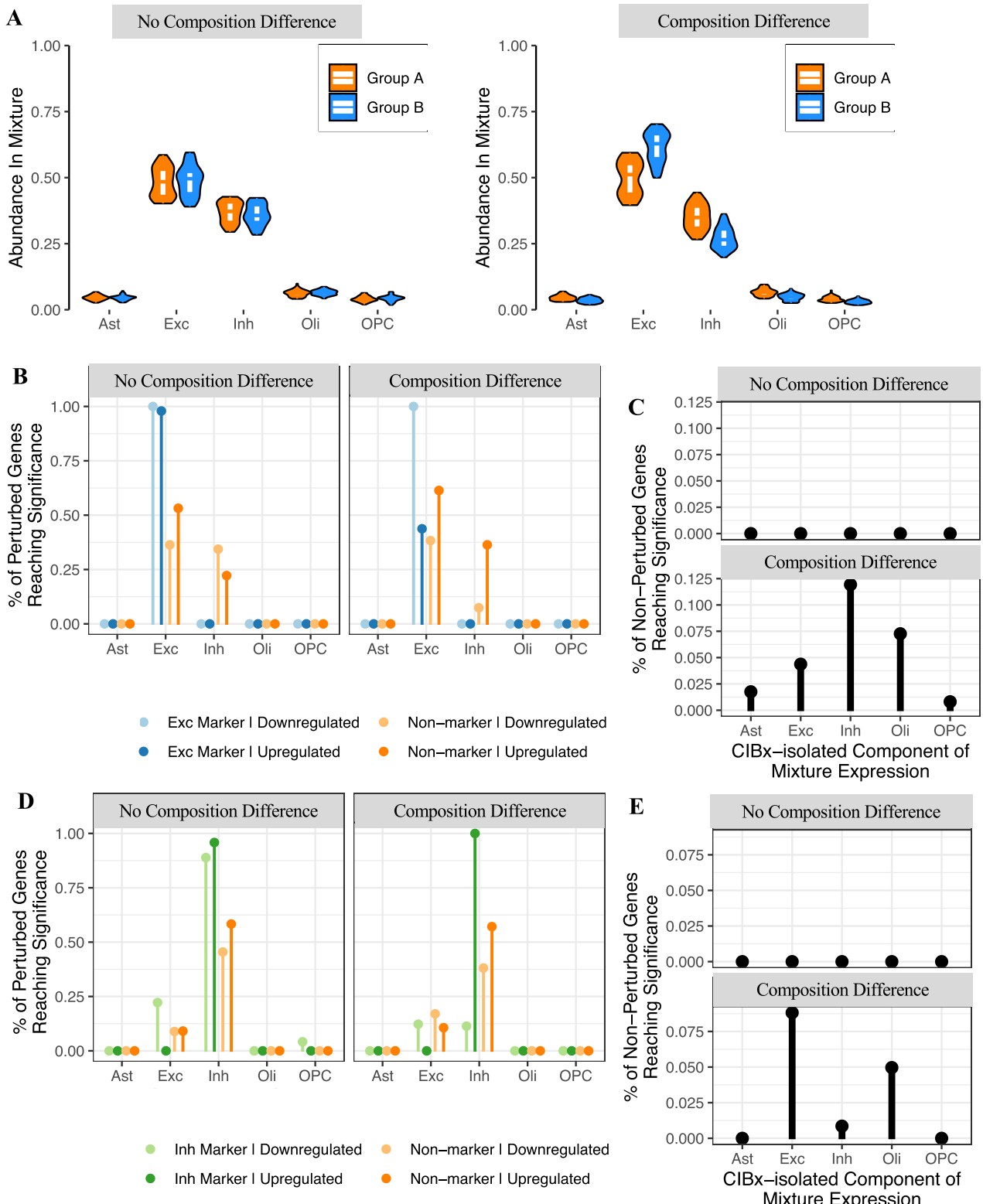

signatures derived from cortex, reasoning that this approach will average-out inter-individual and technical differences, as previously proposed[43].

The accuracy of composition estimates was first evaluated on cortical samples using goodness-of-fit, i.e. the Pearson correlation between measured gene expression and reconstructed gene expression values (see the "Methods"). Consistent with the results on simulated data, we found that cell-type signature data had a stronger impact on accuracy than the choice of algorithm (Supplementary Fig. 33). Although there was some variation between the two datasets, the CA and MultiBrain signatures performed consistently well, while the cultured-cell-derived F5 and the single-nucleus LK signatures performed worst (Fig. 6A, B). Cerebellar samples showed lower goodness of fit than cortical samples in both datasets (Supplementary Figs. 34, 35), consistent with the fact that all cell-type signatures were derived from

**Fig. 5 Cell-type-specific differential expression analysis using CIBERSORTx. A** Violin plots showing the composition distribution of two simulated datasets. Each point represents the proportion of a given cell-type in each dataset. The width of the violin indicates point density, with the top, middle, and bottom of the white overlay box marking the 75th, 50th, and 25th percentiles, respectively. *Left*: simulated data without a composition difference between the two groups. *Right*: simulated data with a composition difference between the two groups. Each group contained 50 samples. **B** Gene expression was perturbed 1.5-fold in Group B in excitatory neurons for 100 non-marker genes plus 100 markers of the given cell-type. CIBERSORTx was used to extract cell-type-specific expression (see the "Methods" section), with a linear model then run to assess differential expression in each cell-type. The plot displays the fraction of the true perturbed genes with an FDR < 0.05. Note that the fraction was calculated using only the subset of perturbed genes which were detected in the given cell-type. **C** False positive rate across the different cell-types when expression was perturbed in excitatory neurons with or without an additional composition difference. **D** As per **B** with the expression perturbation in inhibitory neurons. **E** As per **C** with the expression perturbation in inhibitory neurons. Ast astrocytes, Exc excitatory neurons, Inh inhibitory neurons, Oli oligodendrocytes, OPC oligodendrocyte precursor cells.

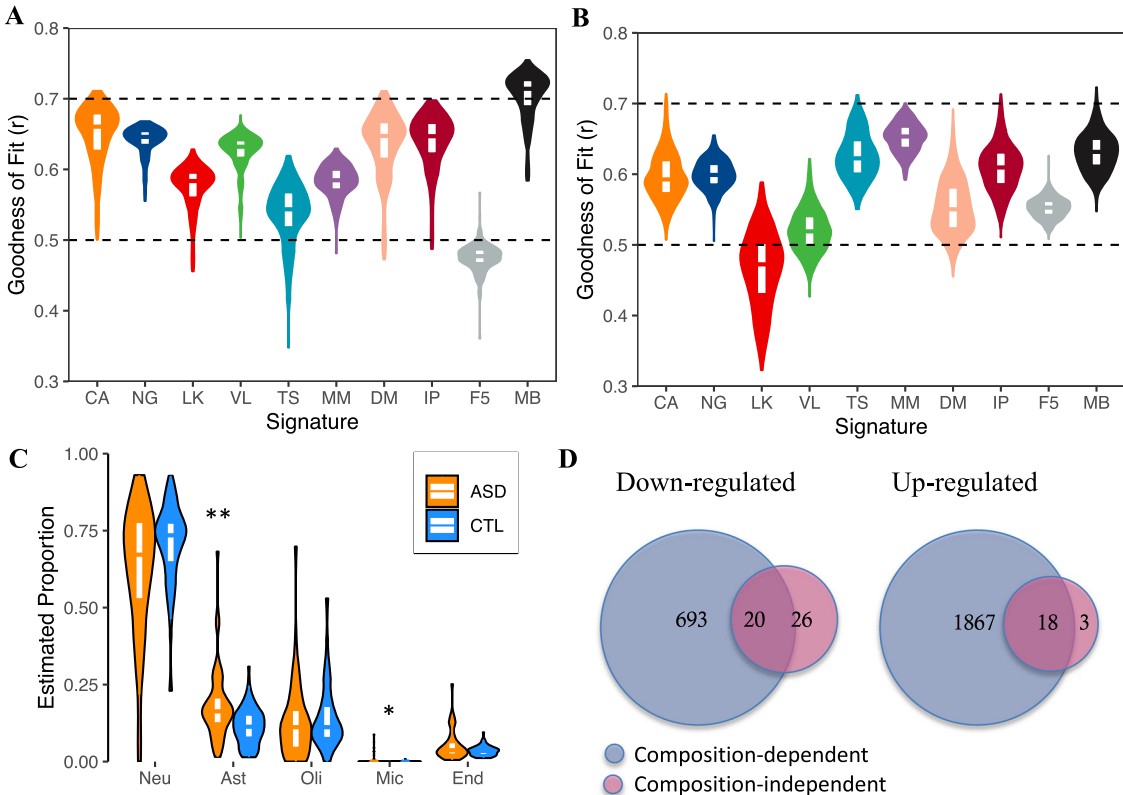

**Fig. 6 Cell-type composition estimates in large-scale human brain transcriptome data. A** and **B** Violin plots of goodness of fit across signatures in cortex samples from (**A**) the GTEx consortium and (**B**) Parikshak et al. Deconvolution was performed using CIBERSORT. The width of the violin indicates point density, with the top, middle, and bottom of the white overlay box marking the 75th, 50th, and 25th percentiles, respectively. *Dotted horizontal lines:* $y = 0.5$ and $y = 0.7$. Signatures in the x-axis are as follows: VL Velmeshev, NG Nagy, CA Human Cell Atlas, LK Lake, TS Tasic, F5 FANTOM5, IP Immuno-paned, MM mouse immuno-panned, DM Darmanis, MB MultiBrain. See the "Methods" section for further details about signatures. **C** Violin plots of composition estimates in ASD ($n = 43$) and Control ($n = 63$) cortical samples from Parikshak et al.[32]. Deconvolution was performed using CIBERSORT and the MultiBrain signature. ASD autism spectrum disorder, CTL control. \*$p < 0.01$. \*\*$p < 0.001$. p-values were calculated using a two-sided Wilcoxon test, without multiple-testing adjustment. The width of the violin indicates point density, with the top, middle, and bottom of the white overlay box marking the 75th, 50th, and 25th percentiles, respectively. **D** Venn diagrams of the overlap between composition-dependent and composition-independent differentially expressed genes between ASD and CTL samples. Differential expression was performed using DESeq2. The composition-independent model included astrocyte, oligodendrocyte, and microglial proportions as covariates.

cerebral cortex. When the biological and technical differences between the bulk data and cell-type signatures are eliminated, as is the case of our in silico mixtures of single-cell data, goodness-of-fit averaged ~0.95 (Supplementary Fig. 36). Further details on the deconvolution results of the GTEx and Parikshak et al. data (correlation and absolute values of cell-type abundance estimates) are included in Supplementary Note.

While the important role of signature data has been previously reported[43], here we discovered the effect of biological factors that affect its influence on accuracy, in particular in vitro culturing and brain region. Since in vitro culturing may be relevant to other tissues as well, we investigated whether using cultured-cell-derived

or tissue-derived signature data influences the accuracy of deconvolution for two additional tissues in GTEx: pancreas and heart. We found that deconvolution accuracy for left heart ventricle and arterial appendage samples was significantly reduced when using signature data from cultured cells, while the deconvolution accuracy for pancreas data was mildly reduced (Supplementary Fig. 37). These data suggest that the influence of biological factors on cell-type signature data vary across tissues, indicating that tissue-specific benchmarking of deconvolution approaches is warranted.

Finally, we applied the results of the cell-type composition analyses to get further insights into genes differentially expressed

in brain tissue samples from Autism Spectrum Disorder (ASD) cases[32]. Cell-type proportion estimates (CIBERSORT/Multi-Brain), showed significantly higher astrocyte proportions in ASD cortex samples compared to controls (difference in means: 7.2%, $p = 0.0002$, Wilcoxon rank sum test; Fig. 6C). This result recapitulates recent single-nucleus data from ASD brain validated by immunohistochemistry[40], which showed higher proportion of astrocytes in ASD cortex samples. There were also significantly higher proportions of microglia (0.7%, $p = 0.003$; Wilcoxon rank sum test), although the overall abundance of microglia was low.

We next carried out differential expression analyses either without correction for cellular composition (composition-dependent; CD) or including cell-type proportion estimates from CIBERSORT/MultiBrain in the model (composition-independent; CI); Methods. Astrocyte, oligodendrocyte, and microglial proportions were included as covariates. CD analyses identified 713 down-regulated and 1885 up-regulated genes (Fig. 6D). In contrast, when correcting for composition estimates in our CI analyses, we identified only 46 down-regulated and 21 up-regulated genes (Fig. 6D). Of these, 20 down-regulated and 18 up-regulated genes overlapped between CI and CD analyses (Fig. 6D). Thus, 26 down-regulated and 3 up-regulated genes were uncovered by the CI analysis, and have not previously been reported as differentially expressed in ASD[32]. Conversely, 693 down-regulated and 1867 up-regulated genes were identified in the CD analysis only, and thus likely reflect differences in cellular composition between the ASD and control samples, rather than gene expression dysregulation (Supplementary Data 4). The CD upregulated genes were enriched for immune and inflammatory genes (Supplementary Data 4) as well as astrocyte and microglial markers ($p = 2.5 \times 10^{-11}$ and $4.7 \times 10^{-36}$, respectively), consistent with their higher proportions in ASD samples. Notably, one of the top up-regulated CI analysis-specific genes, *CXXC4*, which encodes a protein involved in Wnt signalling, has also been identified as upregulated in ASD CTX layer 4 neurons by single-nucleus RNA-seq[40]. In addition, *CXXC4* was identified as the top associated gene in a GWAS meta-analysis of schizophrenia and ASD[53]. These data indicate that correction for cellular composition can identify disease-relevant gene expression changes.

## Discussion

Here, we began to address the question of tissue-specificity in transcriptome deconvolution, by carrying out a comprehensive benchmarking of deconvolution methods on brain transcriptome data. We assessed eight deconvolution methods, as well as multiple parameters of deconvolution: the biological and technical properties of the cell-type signature data; the effect of deconvolving brain cell sub-types; the effect of missing cell-types in the signature data; and the effect of nuclear-enriched or depleted transcripts on deconvolution using snRNA-seq signatures. We also investigated how effectively cell-type composition differences can be corrected in DE analyses.

It has previously been shown that cell-type signature data has a strong effect on deconvolution accuracy[43,54]. In blood, the microarray platform was the main factor driving differences between cell-type signature datasets[43]. For deconvolution of solid tumours, accurate estimation of immune cell-type composition required tumour-derived cell-type signatures, rather than blood-derived signatures[54]. For brain transcriptomes, we found that cell-type signature data had a stronger impact than the choice of method in all cases studied: simulated single-cell mixture data, RNA mixtures of known composition, immuno-panned cells, and large-scale post-mortem transcriptome data. We also found that for brain transcriptomes, biological factors outweighed technical factors, and among biological factors in vitro culturing

(Supplementary Figs. 33–35) and brain region (cortex vs. cerebellum) (Supplementary Figs. 33–35) had the strongest impact. In vitro culturing also affected the performance of deconvolution for other tissues (heart and pancreas), but to different extents (Supplementary Fig. 37), highlighting the importance of tissue-specific benchmarking.

We found that snRNA-seq derived cell-type signatures performed well, particularly the Human Cell Atlas data (CA), which has high-coverage, while low sequencing depth (LK) led to reduced accuracy. Removing compartment-specific genes from the snRNA-seq signatures improved the deconvolution accuracy (Supplementary Fig. 25).

Another factor known to influence deconvolution accuracy is the absence of cell-types present in mixtures from the signature data[21,22]. Consistent with previous results[21,22], we found that if an abundant brain cell-type was missing from the signature data, the deconvolution accuracy was reduced, particularly for cell-types highly correlated with the missing cell-type (Supplementary Fig. 19). The absence of a lowly abundant cell-type, such as microglia and endothelia, had a minimal impact on deconvolution accuracy, suggesting that signature datasets missing these cell-types can be used in deconvolution of brain data.

Since neuronal sub-types are highly similar in gene expression profiles, we investigated how different deconvolution methods handled collinearity in brain transcriptome data. We found that CIBERSORT best handled collinearity, and deconvolution of brain cell sub-types was accurate provided that they were not lowly abundant (<2%) or highly collinear with other cell-types ($rho > 0.95$); (Supplementary Figs. 14, 18).

It was previously shown that semi-supervised and unsupervised complete deconvolution methods underperform relative to supervised (i.e. partial) deconvolution methods[21,22]. Our results support these observations, and we further determine that the range of cell-type composition across samples in the bulk dataset is a major factor influencing the performance of complete deconvolution methods (Fig. 3, Supplementary Figs. 27–30).

When assessing the interplay between cellular composition and DE analyses, we found that false-positives are induced in DE analyses by as low as 5–10% difference in cell-type composition (Fig. 5A). Inclusion of cell-type composition estimates as covariates effectively eliminated composition-induced false-positive genes, and restored discriminatory power for gene expression differences of 2-fold when cell-type composition differences were up to ~25% (Fig. 5C, D, Supplementary Fig. 31).

The deconvolution of large GTEx data and PsychENCODE data showed that the best-performing signature may differ across datasets, and thus it is worth assessing goodness-of-fit for multiple signatures when deconvolving brain gene expression data. Notably, in both datasets, and across all deconvolution methods, there was a wide range of estimated cell-type proportions in any given brain region (Supplementary Note). This is consistent with data from the PsychENCODE consortium[15], which used an NLS-based approach (similar to the one implemented in DeconRNASeq) and reported a similarly wide range of proportion of neurons across 1867 dorsolateral prefrontal cortex samples: 2–54% (http://resource.psychencode.org, PEC_DER-24_Cell-Fractions-Normalised). Such a wide range is also observed in brain methylome deconvolution[55] (0–50%) and likely reflects technical variability in dissection rather than biological inter-individual variability.

Overall, for deconvolution of brain transcriptome data we recommend that (a) CIBERSORT[18] and either dtangle[35,56] or MuSiC[34] are good choices of methods (b) cell-type signature data should be well matched to the bulk samples, in terms of in vitro culture state and brain region, (c) cellular sub-types should only be included in deconvolution if they are neither lowly-abundant

nor highly correlated with other cell-types/sub-types, (d) when using snRNA-seq based signatures, removal of nuclear-specific genes (Supplementary Data 1) from the signature should be considered, (e) only attempt to use reference-free deconvolution methods if the bulk dataset is known to have a wide range of cell-type compositions.

To facilitate the choice of cell-type signature data, we developed a web tool which implements the best performing algorithms and all the cell-type signatures (Supplementary Data 5), as well as calculation of goodness-of-fit, in a user-friendly format, available at: https://voineagulab.shinyapps.io/BrainDeconvShiny/.

## Methods

**Statement of ethics**. Post-mortem human brain tissue transcriptome analyses were undertaken under a protocol approved by the University of Western Australia Human Research Ethics Committee (RA/4/20/6394). De-identified post-mortem samples were obtained from the NIH NeuroBioBank. Informed consent from donors, next-of-kin and/or legally authorised representative was requested and documented by trained individuals, including consent for sample collection, usage by approved researchers, and access to antenatal medical records. All policies and procedures were approved by the collecting institutions' respective Institutional Review Boards and additional oversight committees.

### Datasets accessed and pre-processing

*Bulk tissue RNA-seq resources*. Bulk brain gene expression data from Parikshak et al.[32] were obtained from Github (https://github.com/dhglab/Genome-wide-changes-in-lncRNA-alternative-splicing-and-cortical-patterning-in-autism/releases). Exon-level count data was obtained for 251 post-mortem samples (rRNA-depleted), including frontal cortex, temporal cortex, and cerebellar vermis samples from 48 ASD and 49 control individuals, aged 2–67 (Supplementary Data 6; see Parikshak et al.[32] for complete metadata).

Gene-level normalised data was generated by aggregating exon counts followed by reads per kilobase per million reads (RPKM) normalisation using the total exonic length of each gene (Ensembl V19 (hg19) assembly). A minimum expression threshold was then set at >1 RPKM in at least 40 samples (i.e., half of the number of samples in the least-represented region). Outlier samples removed in the Parikshak et al. study were also removed from our analyses, leaving 121 ASD (43 frontal cortex, 39 temporal cortex, 39 cerebellum) and 126 control (45 frontal cortex, 36 temporal cortex, 45 cerebellum) samples.

Bulk brain gene expression data from GTEx[14] were obtained as gene-level read counts from the 2016-01-05 release (V7) at https://gtexportal.org/home/datasets. Counts were RPKM normalised as above. A minimum expression threshold was set at >1 RPKM in at least 88 samples (i.e. the number of samples in the least-represented brain region).

Bulk gene expression data from GTEx[14] for pancreas ($n = 268$) and heart ($n = 310$ and 417 atrial appendage and left ventricle, respectively) were processed as per the GTEx brain samples, except the pancreas samples were normalised to the level of transcripts-per-million (TPM).

*Brain cell-type-specific gene expression datasets and generation of cell-type signatures*. Information about final expression values and samples used are available in Supplementary Data 5 and 6, respectively. Metrics of signature similarity are presented in Supplementary Figs. 38 and 39.

F5 (FANTOM5): Cap Analysis of Gene Expression (CAGE) data for robust CAGE peaks was obtained from the FANTOM5 consortium: http://fantom.gsc.riken.jp/5/data/[48]. Tag-per-million normalised CAGE peak expression levels were aggregated by sum at gene level. Data from cultured neuron ($n = 3$) and astrocyte ($n = 3$) samples were averaged to generate the F5 neuron and astrocyte signatures. A minimum expression threshold was set at >1 tag-per-million in at least one cell-type.

IP (immuno-purified): RNA-seq data from cells immunopurified from human adult brain tissue extracted during surgery were obtained from Zhang et al.[42]. FPKM-level data were accessed from Table S4 of Zhang et al. for neurons ($n = 1$), astrocytes ($n = 12$), oligodendrocytes ($n = 5$), microglia ($n = 3$), endothelia ($n = 2$). Cell-types derived from foetal brain were excluded (i.e., foetal astrocytes). Samples of the same cell-type were averaged to generate the IP signature. A minimum expression threshold was set at >1 FPKM in at least one of the five cell-types in the final signature matrix.

MM (Mus musculus): RNA-seq data from immunopurified mouse brain tissue was obtained from Zhang et al.[47]. FPKM-level data were accessed from https://web.stanford.edu/group/barres_lab/brain_rnaseq.html, in which biological replicates of cell-type transcriptomes (neurons, astrocytes, oligodendrocytes, microglia, and endothelia) were already aggregated across samples. Mouse genes were mapped to human orthologues using Gene ID homology information from http://www.informatics.jax.org/downloads/reports/HOM_MouseHumanSequence.rpt. Expression data from oligodendrocyte precursors and newly formed oligodendrocytes

were excluded. A minimum expression threshold was set at >1 FPKM in at least one of the five cell-types in the final signature matrix.

DM (Darmanis): Human brain single-cell gene expression data from the middle temporal gyrus generated by Darmanis et al.[13] were downloaded as count-level data from https://github.com/VCCRI/CIDR-examples/tree/master/Brain[57]. To generate the DM signature, RPKM or counts-per-million (CPM) expression was averaged across samples of each cell-type (i.e. astrocyte ($n = 62$), neuron (161), microglia (16), mature oligodendrocytes (38), oligodendrocyte precursor cells (OPCs) (18), or endothelia (20). Cell-types derived from foetal brain (quiescent neurons and replicating neurons) were excluded. A minimum expression threshold was set at >1 RPKM or CPM in at least one cell-type in the final signature matrix.

LK (Lake): Gene expression data for 10,319 human adult frontal cortex nuclei were accessed from Lake et al.[45]. Seurat[58] was used to pre-process raw count expression data, removing nuclei with (1) fewer than 1000 counts, or (2) 200 expressed genes, or (3) >5% of counts attributed to mitochondrial genes, or (4) a number of reads >99.5th percentile of its dataset. Only 3930 nuclei passed these QC criteria. To generate the LK signature, RPKM or CPM values were averaged across nuclei of each cell-type: astrocytes (97), excitatory neurons (2611), inhibitory neurons (1051), oligodendrocytes (96), OPCs (46), and microglia (22). An expression profile for neurons was also generated, as the average of all excitatory and inhibitory nuclei. A minimum expression threshold of >1 RPKM or CPM in at least one cell-type was required. Note that endothelia were excluded for having fewer than 10 nuclei (7).

VL (Velmeshev): 10X Chromium for single-nucleus data from the post-mortem adult human brain were accessed Velmeshev et al.[40]. Only nuclei from control prefrontal cortex samples were included. Seurat processing, cell-type aggregation, and thresholding were performed as described above in LK. After filtering, 24,556 nuclei remained, classified as astrocytes (2229), excitatory neurons (9718), inhibitory neurons (4238), oligodendrocytes (4721), OPCs (2677), microglia (450), and endothelia (523).

CA (Cell Atlas): Count-level exon expression data for NeuN+ sorted adult nuclei from the middle temporal gyrus were acquired from the Human Cell Atlas[41]. Seurat processing, cell-type aggregation, and thresholding were performed as described above in LK. After filtering, 15,524 nuclei remained, classified as astrocytes (291), excitatory neurons (10492), inhibitory neurons (4118), oligodendrocytes (313), OPCs (238), microglia (63). Endothelia were excluded for having fewer than 10 representatives (9).

NG (Nagy): 10X Chromium single-nucleus expression data from the adult human post-mortem human prefrontal cortex were accessed from Nagy et al.[44]. Only nuclei from control samples were included. Seurat processing, cell-type aggregation, and thresholding were performed as described above in LK. After filtering, 23,168 nuclei remained, classified as astrocytes (1195), excitatory neurons (14,624), inhibitory neurons (5940), oligodendrocytes (757), OPCs (505), microglia (85), and endothelia (62).

TS (Tasic): Exon-level SmartSeq2 single-cell expression data from the adult mouse cortex were accessed from Tasic et al.[46]. Only cells from the Anterior Lateral Motor Cortex were included. Further, cells labelled by the authors as low quality or with no class were excluded. Seurat processing, cell-type aggregation, and thresholding were performed as described above in LK. After filtering, 8075 nuclei remained, classified as astrocytes (195), excitatory neurons (3851), inhibitory neurons (3767), oligodendrocytes (69), OPCs (24), microglia (80), and endothelia (89).

MB (MultiBrain): this composite signature was generated by quantile-normalising and averaging the RPKM-level expression of the CA, IP, DM, NG, and VL signatures for five cell-types (neurons, astrocytes, oligodendrocytes, microglia, and endothelia). All signatures are cortical in origin but represent a range of purification protocols (scRNA-seq by SmartSeq (DM), snRNA-seq by 10X (VL, NG), snRNA-seq by SmartSeq (CA), and immuno-panning (IP)).

*Pancreas cell-type-specific gene expression datasets and generation of cell-type signatures*. Cell-type-specific RNA-seq data from pancreas alpha and beta cells were obtained from three studies as described below. For each dataset, genes were excluded if they were not protein-coding, or if they were expressed at <1 TPM in both cell-types.

EN (Enge): count-level expression data for single-cells from freshly isolated, FACS-sorted human pancreas were acquired from Enge et al.[59]. Data were normalised to the level of transcripts-per-million (TPM), using the total exonic length of each gene per the Ensembl V19 (hg19) assembly. The expression signature of alpha and beta cells was generated as the average of 998 alpha and 348 beta cells.

BL (Blodgett): TPM-level expression data for bulk RNA-seq from freshly-isolated, FACS-sorted alpha and beta cells from human pancreas were acquired from Blodgett et al.[11]. The expression signature of alpha and beta cells was generated as the average of 7 adult alpha-cell and 7 adult beta-cell bulk RNA-seq samples.

FS and FG (Furuyama): count-level expression data for human pancreas alpha and beta cells were acquired from Furuyama et al.[12]. After TPM normalisation, the FS (Furuyama Sorted) signature was constructed from freshly-isolated, FACS sorted alpha and beta cells (average of 5 replicates each), while the FG (Furuyama GFP) signature consists of isolated alpha and beta cells subjected to 1-week of

culturing. These cells had been transduced with a GFP expression vector for imaging purposes[12] (average of 4 and 6 replicates, respectively).

*Heart cell-type-specific gene expression datasets and generation of cell-type signatures.* Cell-type-specific RNA-seq data from heart were accessed from three publicly available datasets, containing cardiomyocytes (CM), cardiac endothelia (EC), cardiac fibroblasts (FC), and smooth muscle cells (SMC). For each dataset genes were excluded if they were not protein-coding, or if they were expressed at <1 RPKM across all four cell-types.

F5 (FANTOM5): Cap Analysis of Gene Expression (CAGE) data for robust CAGE peaks was obtained from the FANTOM5 consortium: http://fantom.gsc.riken.jp/5/data/[48]. Tag-per-million normalised CAGE peak expression levels were aggregated by sum at gene level. $n = 3, 4, 6,$ and 3 for CM, EC, FC, and SMC, respectively.

EN (ENCODE): FPKM-level RNA-seq data for cultured primary cells were accessed from the ENCODE consortium[60]; $n = 2$ for all cell-types.

SC (Single-cell): single-cell RNA-seq data from freshly isolated tissue samples were accessed from Wang et al.[61] (GSE109816). Only left atrial samples were used. Cell-type-specific expression was generated as the average RPKM of all cells in each classification. $n = 1934, 1111, 257,$ and 427 for CM, EC, FC, and SMC, respectively.

## RNA-seq data generated in the present study and data pre-processing

*RNA mixture experiment.* Total RNA was extracted from human primary astrocytes and from neurons derived from human foetal neural progenitors.

Human primary astrocytes (Lonza, #CC-2565) stably expressing GFP from pCMV6-AC-GFP had been generated by selection with G418 (Thermo Fisher Scientific, #10231027) at 800 µg/ml. Cells were cultured in RPMI GlutaMAX™ (Thermo Fisher Scientific, #35050061) supplemented with 10% foetal bovine serum, 1% streptomycin (10,000 µg/ml), 1% penicillin (10,000 units/ml) and 1% Fungizone (2.5 µg/ml) and seeded into six-well tissue culture plates at a density of $0.5 \times 10^6$ cells 24 h prior to RNA extraction. Total RNA was extracted using TRIzol® reagent and a Qiagen miRNeasy kit and treated with 1 µl DNase I (Thermo Fisher Scientific, #AM2238) per 10 µg of RNA.

RNA from differentiated neurons[62] was kindly provided by Dr. Brent Fogel (UCLA). Neurons were differentiated from primary human foetal neural progenitors stably transfected with pLRC-GFP by culturing for 2 weeks in the presence of 1% foetal calf serum 500 ng/mL all trans-retinoic acid[62]. RNA extraction was carried out using a Qiagen miRNeasy kit, with on-column DNase digestion.

RNA mixtures were generated by mixing neuronal and astrocyte RNA in mass ratios of 40:60, 45:55, 50:50 neuron:astrocyte ($n = 1$ for each ratio). In addition, a pure neuronal RNA sample and pure astrocyte RNA samples ($n = 3$) were also included (Supplementary Data 7).

Library preparation using the Illumina TruSeq Stranded kit (http://www.illumina.com/products/truseq_stranded_total_rna_library_prep_kit.html) and sequencing on a NextSeq 500 Illumina sequencer were carried out at the UNSW Ramaciotti Centre for Genomics, generating 75 bp paired-end reads (Supplementary Data 7). Sequencing reads were mapped to the human genome (hg19) using STAR v2.5.2b[63] with the following parameters: --outSJfilterOverhangMin 5 5 5 5 --alignSJoverhangMin 5 --alignSJDBoverhangMin 5 --outFilterMultimapNmax 1 --outFilterScoreMin 1 --outFilterMatchNmin 1 --outFilterMismatchNmax 2 --chimSegmentMin 15 --chimScoreMin 15 --chimScoreSeparation 10 --chimJunctionOverhangMin 5.

Gene counts for GENCODE V19 annotated genes were obtained from the STAR output and RPKM-normalised. Data from the three pure astrocyte replicates were averaged. The signature data consisted of the pure neuronal and pure astrocyte samples, thresholded for a minimum of 1 RPKM in at least one of the two cell-types. All five samples were used as input mixtures for deconvolution, with genes expressed at <2 RPKM in all five samples filtered out. Analyses from these data can be found in Supplementary Figs. 7, 26F, and 28.

*Bulk RNA-seq data generated from brain tissue.* De-identified post-mortem samples were accessed from the NIH NeuroBioBank, and included frontal cortex samples (BA9/10) from 2 control, 2 ASD, and 1 Fragile-X premutation carrier individuals. For each brain sample, frozen tissue was pulverised using a CellCrusher (https://cellcrusher.com/) and the tissue was then divided for nuclear RNA extraction and RNA extraction from bulk tissue.

To isolate nuclei, around 30 mg of tissue was lysed in 2.5 ml lysis buffer (10 mM Tris–HCl, 3 mM MgCl₂, 10 mM NaCl, 0.005% NP40) for 17 min on ice. After lysis, 2.5 ml of ice-cold PBS was added to the sample and tissue was homogenised using a Pasteur pipette until no large chunks were visible. Tissue was then filtered through a 30 µm strainer and centrifuged at $500 \times g$ for 5 min at 4 °C. Supernatant was removed and the pellet was resuspended in 400 µl PBS with 1% BSA and DAPI. DAPI-positive singlet nuclei were sorted using a BD Influx with a 70 µm nozzle at 20 PSI to collect ~100,000 nuclei per sample.

To extract RNA, the Qiagen mini RNA prep kit was used following the manufacturer's instructions, including a DNase treatment step. From sorted nuclei, RNA was extracted by a hot Trizol extraction method. Nuclei were washed in PBS and resuspended in Trizol at 65 °C and incubated on a shaker at 1300 rpm for 15 min. RNA was enriched using a guanidinium HCl buffer and silica-coated

magnetic beads with a DNAse I treatment step. RNA amounts and quality were assessed on a TapeStation using RNA Screen Tape (Agilent), and 20–100 ng of total RNA was used per replicate to generate RNA-seq libraries. ERCC ExFold RNA Spike-In mixes (Thermo Scientific) were added as internal control. Libraries were prepared using the TruSeq Stranded mRNA library prep kit (Illumina), using TruSeq RNA unique dual index adapters. Libraries were quantified by qPCR on a CFX96/C1000 cycler (Bio-Rad) and sequenced on a NovaSeq 6000 (Illumina) for $2 \times 53$ bp as paired-end, generating around 25M reads per sample.

Sequencing reads were mapped to the human genome (hg38) using STAR v2.5.2b[63] with the following parameters: --outSJfilterOverhangMin 15 15 15 15 --alignSJoverhangMin 15 --alignSJDBoverhangMin 15 --outFilterMultimapNmax 1 --outFilterScoreMin 1 --outFilterMatchNmin 1 --outFilterMismatchNmax 2 --chimSegmentMin 15 --chimScoreMin 15 --chimScoreSeparation 10 --chimJunctionOverhangMin 15 --bamRemoveDuplicatesType UniqueIdenticalNotMulti. Note that nuclear samples were mapped to a pre-mRNA hg38 transcriptome.

Gene counts for GENCODE V19 annotated genes were obtained from the STAR output and RPKM-normalised.

Nuclear enrichment was confirmed using the expression of the nuclear-specific transcript MALAT1 (22.1-fold enrichment in nuclear samples, $p = 6.7 \times 10^{-5}$, $t$-test; Supplementary Data 7).

*Single-nucleus RNA-seq data generated from bulk brain tissue.* snRNA-seq data were generated from the same five brain samples described in the previous section, but from a different portion of the dissection.

To isolate nuclei, around 30 mg of tissue was lysed in 400 µl of lysis buffer (10 mM Tris–HCl, 3 mM MgCl₂, 10 mM NaCl, 0.005% NP40) in 1.5 ml tubes and broken down with a pellet pestle. Tissue was dissociated by passing through a polished silanized Pasteur pipette 3–4 times, then incubated on ice for 10 min. Dissociation was repeated at 5 and 10 min. After incubation, the dissociated tissue was added to 2.5 ml of wash buffer in a 15 ml falcon tube. The sample was then passed through a 30 µm strainer into a 50 ml falcon tube and centrifuged for 5 min at $500 \times g$ at 4 °C in a swinging bucket centrifuge. Following centrifugation, the supernatant was removed and sample resuspended in 100 µl of wash buffer (PBS with 1% BSA) for every 30 mg of tissue used. Using only 100 µl of the resuspended sample, 180 µl of a 1.8 M sucrose solution (made with Sigma Nuclei Pure Prep kit) was added and homogenised using a P1000 pipette. In a 2 ml Eppendorf tube, 1 ml of a 1.3 M sucrose solution with 1% BSA was placed. 280 µl of the nuclei suspension mixed with sucrose was slowly layered on top of the 1.3 M sucrose solution. The sucrose gradient was centrifuged for 10 min at $3000 \times g$ at 4 °C in a swinging bucket centrifuge. After centrifugation, the debris from the top of the sucrose gradient were removed by soaking a Kimwipe from the top of the tube, slowly lowering it together with the sinking meniscus until a volume of <100 µl remained, which was removed with a pipette. Nuclei were resuspended in 20–50 µl of wash buffer and 10 µl of the suspension was stained with Trypan Blue to count for concentration.

The 10X Genomics 3′ v2 and v3 single cell expression kit was used to generate single nuclei RNA-seq libraries. Using 16,000 nuclei in total to aim for 10,000 nuclei recovery, the standard protocol was used according to manufacturer instruction with the following alterations: 17 PCR cycles in total for cDNA amplification and 13 PCR cycles in total for library amplification. Libraries were then sequenced on a NovaSeq 6000 (Illumina) generating around 100M reads per sample.

Cell Ranger version 2.1.0 was used to process raw sequencing data. Here, a pre-mRNA transcriptome was built using the Cell Ranger mkref command and default parameters starting with the refdata-cellranger-GRCh38-1.2.0 transcriptome as per the instructions provided by 10X Genomics. Reads were demultiplexed by sample index using the Cell Ranger mkfastq command. Fastq files were aligned to the custom transcriptome, cell barcodes were demultiplexed, and UMIs corresponding to genes were counted using the Cell Ranger count command using default parameters.

Cell Ranger output was pre-processed using Seurat v3[58]. Filtering-out criteria for nuclei: <500 counts, or <200 expressed genes, or >20% of counts attributed to mitochondrial genes, or total number of reads in the top 99.5th percentile of its dataset. UMI counts were log2-transformed and normalised for library size and mitochondrial percentage, and finally scaled. Nuclei from all individuals were then integrated using canonical correlation analysis in Seurat, setting the numbers of dimensions to be 30.

After retransforming and renormalising data, clustering was performed using tSNE[64] in Seurat on the top 35 principal components of the 2000 most variable genes, with the resolution parameter set to 1.5 (Supplementary Fig. 40). Clusters were annotated using SingleR[65] to transfer cell-type annotation labels from the NG signature.

A separate cell-type-specific signature was generated for each of the five individuals. This was calculated as the average RPKM of each individual's cells within each cluster. Only cell-types represented in all individuals were used (Neurons, Astrocytes, and Oligodendrocytes).

## Simulated datasets

*Randomly sampled single-nucleus mixtures.* Used in Figs. 1, 2 and 3A, B, and Supplementary Figs. 1, 2–6, 9–24, 26A, C, 27A, B, 27D, E, and 36A, B.

Simulated data was generated separately from the VL[40] and the CA[41] datasets. Seurat v3 was used to pre-process raw count expression data, removing nuclei with (1) fewer than 1000 counts or 200 expressed genes, (2) >5% of counts attributed to mitochondrial genes, or (3) a number of reads >99.5th percentile of its dataset. In addition, cells assigned to a cell-type or cell-subtype with fewer than 200 cells were excluded. Next, the dataset was randomly split into two: half was used to generate cell-type signatures and half for simulated mixture. One hundred mixtures were simulated by summing the counts of 500 randomly sampled single nuclei. Random sampling was performed without replacement.

*Randomly sampled single-cell mixtures*. Used in Fig. 3A, B, and Supplementary Figs. 2, 22, 26E, and 36C.

Randomly sampled single-cell mixtures were generated using single-cells from the Darmanis et al. dataset[13], using a method largely as per the previous section but with three key differences: first, only cells classified as one of neurons, astrocytes, oligodendrocytes, OPCs, microglia, or endothelia were included, without regard for the number of representatives (i.e., non-hybrid cells from adult samples); second, the number of cells aggregated per mixture was only 100, owing to the lower number of total cells (285); and finally, the dataset was not randomly split into two for mixture and signature generation.

We confirmed that the CA-derived and VL-derived simulated single-nucleus mixtures had similar expression distributions to data from bulk brain tissue, whilst DM-derived simulated single-cell mixtures were not zero-inflated unlike single cells (Supplementary Fig. 41).

*Single-nucleus mixtures sampled with a wide range of cell-type compositions*. Used in Fig. 3D–F, and Supplementary Figs. 29 and 30.

One hundred mixtures were simulated using single nuclei from each of the VL and CA datasets. To obtain a defined range of cell-type proportions in the mixture, for each cell-type $j$ we randomly sampled without replacement between 1 and $n_j$ nuclei where $n_j = (n/k)/(s_j/\min(s))$ where $n = 500$, the chosen number of cells per mixture; $k$ is the number of cell-types in each dataset; $s$ is the vector of total library sizes for all $k$ cell-types; $s_j$ is the total library size for cell-type $j$.

If more than 500 total nuclei were randomly sampled by this approach, then a random subset of 500 was kept; conversely, if fewer than 500 nuclei were initially sampled, then additional nuclei were randomly-sampled from any cell-type until 500 was reached. Mixtures were simulated by summing the counts of these single nuclei followed by counts-per-million normalisation.

*Simulated datasets with cell-type composition differences between sample groups*. Used in Fig. 4A and B.

Single-nucleus mixtures for DE analyses were generated using snRNA-seq data from the CA dataset. Nuclei were classified as one of Excitatory, Inhibitory, Oligodendrocyte, OPC, or Astrocyte. Each simulation was created as a dataset of 100 samples, split into groups A and B of 50 samples each. Each sample in group A (the reference group) was generated as the summed expression of randomly selected $n$ excitatory neurons and 500-$n$ non-excitatory cells, where $n$ was a randomly selected integer from [200–300] so that the simulated proportion of excitatory neurons varies between 40% and 60%). Samples in group B (test group) were generated as per group A, except n was sampled from $[200 + k, 300 + k]$ for increased proportions or $[200−k, 300−k]$ for decreased proportions where $k$ varied from 0 to 195 with a step of 5. All sampling was performed without replacement. Differential expression analyses for group B vs. group A were performed on each dataset as described in the "Differential expression" section below.

*Simulated datasets with cell-type composition and gene expression differences between sample groups*. Used in Fig. 4C, D and Supplementary Fig. 31.

The expression of 200 genes was altered by 1.1-, 1.3-, 1.5-, or 2-fold in the above simulated mixtures, in group A samples only, by multiplication of counts prior to CPM normalisation. The 200 genes selected for perturbation included the top 100 excitatory neuron marker genes and 100 randomly selected non-marker genes. Half of each set was randomly assigned to be upregulated or downregulated.

To simulate cell-type-specific expression differences, the expression alteration was introduced only to nuclei from the cell-type-of-interest (i.e. excitatory or inhibitory neurons) prior to aggregation.

*Pseudo-bulk ASD vs. Control simulated data*. Used in Supplementary Fig. 32.

Single-nucleus data from Velmeshev et al.[40] were processed using Seurat[58], removing nuclei that fulfilled any of the following filtering criteria: <1000 total read counts, <200 expressed genes, >5% of counts attributed to mitochondrial genes, total number of reads >99.5th percentile. Only individuals with >100 nuclei in each of astrocytes, OPCs, excitatory neurons, and inhibitory neurons were included (15 ASD and 11 controls). Nuclei from the prefrontal cortex and anterior cingulate cortex were pooled.

Pseudo-bulk mixtures were made by summing the expression counts of all relevant nuclei, and normalising to counts-per-million. Cell-type-specific expression mixtures for astrocytes, OPCs, excitatory neurons, and inhibitory neurons were generated separately from each individual, with an additional cell-type signature made by pooling nuclei across all individuals. A heterogenous

pseudo-bulk sample was generated from each individual by pooling all their nuclei regardless of cell-type label, and used as input for CIBERSORTx.

## Estimation of cellular composition

*Overview of deconvolution methods*. In general, deconvolution methods model gene expression data from a tissue sample (vector $\mathbf{X}$) as the sum of gene expression levels in the cell-types of which it's comprised ("signature" expression matrix, $\mathbf{S}$), weighted by the proportion of each cell-type in the sample (vector $\mathbf{P}$), formalised as $\mathbf{X} \sim \mathbf{SP}$. Deconvolution methods fall into two broad categories—partial and complete—as described below.

Partial or supervised deconvolution[6,18,33,35,66–71] estimates the proportion of cell-types in a sample based on experimentally measured gene expression values from pure cell-types, i.e. determines $\mathbf{P}$ knowing $\mathbf{X}$ and $\mathbf{S}$.

It is worth noting that the signature expression data $\mathbf{S}$ often comes from a different source than the bulk tissue data $\mathbf{X}$, and thus an intrinsic assumption of most partial deconvolution methods is that gene expression in a given cell-type is the same regardless of the source of cells (thus genetic background and environmental conditions including culture conditions are ignored)[1,71]. The most frequently employed methods for partial deconvolution are Non-negative Least Squares (i.e. optimising $\mathbf{X} \sim \mathbf{SP}$ using a least-squares approach where $\mathbf{P}$ should be non-negative) (e.g. DeconRNASeq[33]), and Support Vector Regression (e.g. CIBERSORT[18]).

A simplified version of partial deconvolution consists of calculating an enrichment score, rather than a proportion, for each cell-type (e.g. xCell[19], or BrainInABlender[7]). While this approach is intuitive, it has several limitations: its accuracy is harder to assess (as one cannot calculate error measures or goodness-of-fit), and its biological interpretation is often unclear since the scale of enrichment scores is variable.

In contrast, complete or reference-free/unsupervised deconvolution consists of estimating both the proportion of cell-types and cell-type-specific expression, i.e. determining both $\mathbf{P}$ and $\mathbf{S}$ knowing $\mathbf{X}$[37–39,49,72]. This is an under-determined problem, which requires biologically-motivated constraints.

*Application of deconvolution methods*. Cell-type composition was estimated using four partial deconvolution methods (DeconRNASeq[33], dtangle[35], MuSiC[34], and CIBERSORT[18]), two enrichment methods with in-built signatures (BrainInABlender[7] and xCell[19]), and two complete deconvolution methods: Linseed[37], and a co-expression-based approach proposed by Kelley et al.[5] (referred to as Coex).

All algorithms were run in R v3.6. All data used for deconvolution were RPKM-normalised expression values without log2 transformation[73] unless noted below.

CIBERSORT v1.04 was run using the CIBERSORT R package obtained from https://cibersort.stanford.edu with default parameters.

DeconRNASeq v1.26 was run using the DeconRNASeq Bioconductor R package with default parameters.

MuSiC v0.1.1 was run using the music_prop() function from R package available at https://github.com/xuranw/MuSiC. Raw count data was used as input for both signatures and mixtures. Only single-cell- or single-nucleus-derived signatures were used; their individual cells/nuclei were not aggregated, metadata about the individual-of-origin was included as well as predefined cell-type labels.

dtangle v0.3.1 was run using the dtangle CRAN R package. Cell-type markers were selected as the top 1% of markers using its find_markers() function with method = "diff". Data was log2 transformed with an offset 0.5, as recommended[35].

BrainInABlender v0.9 was run using the R package obtained from https://github.com/hagenaue/BrainInABlender using default parameters. Cell-type signature data built into BrainInABlender is derived from numerous resources of brain cell-type-specific expression, including human data from Darmanis et al.[13], and various mouse datasets (full list in Hagenauer et al., 2018). Both publication-specific indices and an averaged index are generated; we used the averaged index as the enrichment score in all analyses.

xCell v1.1.0 was run using the R package from https://github.com/dviraran/xCell using default parameters with the built-in signature data. Cell-type signature data for neurons and astrocytes are in-built in xCell, and are derived from in vitro cultured data from FANTOM5[48], and ENCODE[60]. xCell generates a "Raw" and a "Transformed" enrichment score; we used the latter as a measure of enrichment.

Coex was carried out by constructing co-expression networks using the *blockwiseModules* function from the WGCNA R package[50,74], with the following parameters: deepSplit = 4, minModuleSize = 150, mergeCutHeight = 0.2, detectCutHeight = 0.9999, corType = "bicor", networkType = "signed", pamStage = FALSE, pamRespectsDendro = TRUE, maxBlockSize = 30,000. The beta power was selected for each network so that the scale-free topology fit $r^2$ was >0.8 and median connectivity < 100. Genes were assigned to the module with the highest kME (correlation with the module eigengene), provided kME > 0.5, and $p < 0.05$ (BH-corrected Student's t-test). Co-expression networks were built on log2-transformed RPKM values, offset by 0.5.

A cell-type module (CTM) was defined as the module most significantly enriched for the top 100 markers of a given cell-type, requiring an enrichment $p$-value < $10^{−5}$ and odds ratio >5.

Enrichment was assessed using a one-sided Fisher's Exact Test. Cell-type markers were defined using the find_markers() function in the dtangle R package

applied to the matching cell-type signature data for simulations, and MB for GTEx and Parikshak. Cell-type enrichment scores were defined as the CTM's eigengene values (i.e., first principal component values of genes included in the CTM), as per Kelley et al.[5].

Linseed v0.99.2 was run using the R package from https://github.com/ctlab/LinSeed. We used a collinearity threshold of $p = 0.01$ to filter genes. Output was transformed to sum-to-one.

We also tested the SVD approach to determine the number of cell-types in the mixture data, which involves looking for the plateau (Supplementary Fig. 26). For the VL-based simulations, with 7 cell-types, the estimated $k$ was >10 for random and 7 for wide-range mixtures. For the CA-based simulations, with 5 cell-types, the estimated $k$ was ~5–7 for random and 5 for wide-range mixtures. For the RNA mixtures which consisted of 2 cell-types, the estimated $k$ was 3. For the DM mixtures, which consisted of 5 cell-types, the estimated $k$ was more than 10. Therefore, we used the known $k$ value for all mixtures.

*Deconvolution parameters for specific datasets.* Parikshak, GTEx, and RNA mixture data were deconvolved using RPKM-normalised signatures and mixtures, while for single-cell and single-nucleus simulated datasets, signatures and mixtures were CPM-normalised, if raw count data was available (otherwise the normalised data available from the original publication was used).

**Assessment of deconvolution accuracy.** For simulated datasets, deconvolution accuracy was assessed by two measures: (i) Pearson correlation between true and estimated proportions and (ii) normalised mean absolute error (nmae) calculated as mean error divided by the mean of true proportions, where error is the per-sample absolute difference between estimate and true proportion. Note that nmae can only be calculated when estimates are bounded between 0 and 1, i.e. are proportions rather than relative enrichment scores like BrainInABlender's or xCell's output.

For datasets without a ground truth, such as bulk brain samples, goodness-of-fit was evaluated as the Pearson correlation for each sample's reconstructed and observed expression, log2-transformed with an offset of +0.5. Here, expression was reconstructed for each gene using the following formula:

$$\sum_{j=1}^{n} s_j \cdot p_j$$

where $j$ denotes a cell-type, $s_j$ is the gene's expression in cell-type $j$ (from the signature matrix), $p_j$ is the estimated proportion of cell-type $j$ in the sample, and $n$ is the number of cell-types. Note observed expression and cell-type signature data were quantile normalised prior to reconstruction.

**Differential expression (DE) analyses**

*DE analyses in simulated bulk data.* DE between group A and group B in simulated single-nucleus mixtures was assessed using either a linear model on log2-transformed CPM values offset by +0.5 as implemented in the lm function in R, or a generalised linear model implemented in DESeq2[51] on count data.

Excitatory neuron proportions were included as covariates in the model either as linear term, a quadratic term, or after conversion to a spline matrix using the bs() function from the R *splines* package, setting degree = 3 and knots at its 25th, 50th, and 75th percentiles.

Multiple testing correction was conducted using the Benjamini–Hochberg approach[75].

Cell-type marker enrichment analyses were performed by one-sided Fisher's exact test for 100 markers per cell-type using the CA cell-type signature. Markers were defined using the find_markers() function from dtangle[35], setting marker_method = "diff". Only simulations where the number of false positive genes was >100 were tested for cell-type enrichment, to ensure adequate power for the test.

Discriminatory power was calculated as the fraction of the 200 true perturbed genes that were in the top 200 most significant genes by $p$-value. No significance threshold was applied to DE $p$-values.

*DE analyses using CIBERSORTx-estimated cell-type-specific gene expression data.* Cell-type-specific expression was first extracted using the high-resolution algorithm of CIBERSORTx[52] webtool at https://cibersortx.stanford.edu/ with default settings and using a subset of 1000 genes, per its computational constraints.

For analyses of simulated data, the CA signature was used, filtered to the 100 perturbed cell-type marker genes, the 100 perturbed non-marker genes, the top 100 markers for each of the four other cell-types, and 400 randomly selected non-marker genes. The resultant cell-type-specific expression data was used for DE analyses, using a linear model on log2 transformed data offset by +0.5. Multiple testing correction was conducted using the Benjamini–Hochberg approach, adjusting for the size of the full transcriptome rather than that of the smaller subset[75].

For the pseudo-bulk dataset of 15 ASD and 11 control samples, the subset of 1000 genes was chosen by performing DE analysis for the effect of diagnosis within each of the corresponding four pure cell-types, and choosing the 250 genes with the

lowest $p$-value. DE was performed using a linear model on log2 transformed data offset by +0.5, adjusting for age, sex, RIN, PMI, and log10 of total UMI count. The same model was applied to the estimated cell-type-specific expression output of CIBERSORTx. DE was only performed on genes which passed CIBERSORTx's quality control, and also had non-zero variance in expression across samples.

*DE analyses of ASD and control samples.* DE was carried out using DESeq2 v1.22.2[51] on count-level expression data. The same samples used by Parikshak et al.[32] for DE were included in our analyses: 106 samples (43 ASD, 63 controls; Supplementary Data 6). Differential expression was carried out using a Wald test with Benjamini–Hochberg correction for multiple testing as implemented in DESeq2[51]. Composition-dependent DE adjusted for the following covariates: Age, Sex, Sequencing Batch, Brain Bank, Region, RIN, and the first two principal components of sequencing metadata, per Parikshak et al.[32]. Composition-independent DE used the same covariates as above, but adding the estimated proportions of astrocytes and any other cell-types not significantly correlated with astrocyte proportions ($p < 0.05$, Pearson correlation test) i.e., oligodendrocytes and microglia (Supplementary Fig. 42), to minimise co-linearity.

*Determination of compartment-specific genes.* Compartment-specific genes were identified using the RNA-seq data generated from bulk brain tissue (5 total RNA and 5 nuclear RNA samples) pre-processed as described above, and log2-transformed with an offset of 0.5. Compartment specific genes were identified as genes DE between groups using a linear model as implemented in the *lm* function in R (absolute fold-change > 1.3 and a Benjamini–Hochberg-adjusted[75] $p$-value < 0.05).

**Other analyses.** Gene ontology (GO) and pathway enrichment analyses were conducted using gProfiler2 v0.2[76] in R, setting exclude_iea = TRUE and all other parameters as default. $p$-values were BH-corrected[75]. Only results from GO, KEGG, Reactome, Human Phenotype, and Wikipathways were reported, with filtering performed after multiple-testing correction.

For all set enrichment analyses, the background was set to the relevant list of all expressed genes.

**Note on cell-type proportions vs. RNA proportions.** Since cell-types differ in their total RNA content, transcriptome deconvolution estimates proportions of RNA from each cell-type, rather than proportions of cells per se[37]. It is important to note that bulk RNA-seq sequences a mixture of RNA molecules (primarily protein coding, after poly-A selection or ribo-depletion), and thus the goal of transcriptome deconvolution is in fact to estimate the proportion of the sequenced RNA molecules coming from a given cell-type (pRNA), rather than the proportion of cells. A priori, pRNA (not pCt) should be relevant for reconstruction of gene expression data, and thus useful as a covariate in differential expression analyses. To test this hypothesis, we deconvolved pseudo-bulk data from the Velmeshev et al. snRNA-seq dataset (10 individuals), where we know both pCt and pRNA (calculated as the proportion of RNA-seq reads from each cell-type), and found that deconvolution estimates perfectly correlate with pRNA but less so with pCt (Supplementary Fig. 43), consistent with previous data[37]. Note that pRNA and pCt are themselves correlated in this dataset ($r = 0.86$). We then assessed goodness-of-fit for these pseudo-bulk data using either pRNA or pCt to reconstruct gene expression. We found that goodness-of-fit was always higher when using pRNA (Supplementary Fig. 43). These data demonstrate that pRNA, the output of transcriptome deconvolution, is the appropriate measure to use for re-constructing gene expression data and thus as a co-variate in DE analyses. For simplicity, we refer to pRNA as "cell-type proportions" throughout the manuscript.

**Reporting summary.** Further information on research design is available in the Nature Research Reporting Summary linked to this article.

## Data availability

The sequencing data generated in this study have been deposited in the GEO database under accession code GSE175772 (Processed signature data can be accessed in Supplementary Data 5.) A website for users to deconvolution their own brain data with the top performing algorithms is implemented at https://voineagulab.shinyapps.io/BrainDeconvShiny/.

The RNA-seq data for bulk brain tissue was accessed from the following two resources: Parikshak et al. (2016) (https://github.com/dhglab/Genome-wide-changes-in-lncRNA-alternative-splicing-and-cortical-patterning-in-autism/releases); and GTEx v7 release (https://gtexportal.org/home/datasets). Bulk pancreas and heart data were accessed from same GTEx resource.

Brain cell-type-specific expression was accessed from the following nine resources: FANTOM5 (http://fantom.gsc.riken.jp/5/data/); Zhang et al. (2016) (https://www.ncbi.nlm.nih.gov/geo/query/acc.cgi?acc=GSE73721); Zhang et al. (2014) (https://web.stanford.edu/group/barres_lab/brain_rnaseq.html); Darmanis et al. (2015) (https://github.com/VCCRI/CIDR-examples/tree/master/Brain); Lake et al. (2018) (https://www.ncbi.nlm.nih.gov/geo/query/acc.cgi?acc=GSE97942); Velmeshev et al. (2019) (https://autism.cells.ucsc.edu/); The Human Cell Atlas (http://portal.brain-map.org/);

Nagy et al. (2020) (https://www.ncbi.nlm.nih.gov/geo/query/acc.cgi?acc=GSE144136); and Tasic et al. (2018) (GSE115746).

Cell-type-specific expression for non-brain tissues were accessed from the following four sources: Enge et al. (2017) (GSE81547); Blodgett et al. (2015) (https://www.ncbi.nlm.nih.gov/geo/query/acc.cgi?acc=GSE67543); Furuyama et al. (2019) (https://www.ncbi.nlm.nih.gov/geo/query/acc.cgi?acc=GSE117454); ENCODE (https://www.encodeproject.org/publication-data/ENCSR590RJC/); and Wang et al. (2020) (https://www.ncbi.nlm.nih.gov/geo/query/acc.cgi?acc=GSE109816).

## Code availability

Data analysis code is available at https://github.com/Voineagulab/BrainCellularComposition.

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

## Acknowledgements

The authors thank Dr. Brent Fogel (UCLA) for the gift of RNA from differentiated neural progenitors, Dr. Akira Gokool (UNSW) for help in the initial stages of the project, and all members of the Voineagu lab for helpful discussions and comments on the paper. This work was supported by an ARC Future Fellowship FT170100359 and a UNSW Scientia Fellowship to I.V., NHMRC project grant APP1138557 to I.V. and R.L., and an Australian Government RTP Ph.D. Scholarship to G.J.S. R.L. was supported by an NHMRC Investigator Grant GNT1178460 and Howard Hughes Medical Institute International Research Scholarship. The bulk and snRNA-seq data from human brain samples was generated on instrumentation supported by the Australian Cancer Research Foundation Centre for Advanced Cancer Genomics and Genomics WA.

## Author contributions

All analysis was performed by G.J.S. with critical input from J.G.B. and I.V. K.W. and U.N. developed BrainDeconvShiny and the associated R package. D.P., R.K.S., and R.L. generated the bulk and snRNA-seq data from human brain samples. The manuscript was written by G.J.S. and I.V. with input from all other authors. I.V. supervised all aspects of the paper.

## Competing interests

The authors declare no competing interests.
