## [Peer Review File · Nature Communications]

Title: Comprehensive evaluation of human brain gene expression deconvolution methodsREVIEWER COMMENTS

Reviewer #2 (Remarks to the Author):

In this paper, Sutton and Voineagu perform a comparison of several bulk gene expression cellular deconvolution methods. Even though the paper has some value in providing side-by-side comparison of existing deconvolution methods, in my opinion it fails to offer novel insights or methodological developments that will be of use in future transcriptomic studies. The current state of the field and outstanding issues the study addresses are not well presented, the datasets used for the main analyses are not up-to-date, and biological assumptions for the analyses and conclusions are oversimplified.

Below are my point-by-point discussion of the paper and major and minor comments.

Major comments:

1) The introduction is rather biased toward listing pros of bulk gene expression deconvolution methods and cons of single-cell transcriptomic methods. This makes it seem as if performing bulk tissue gene expression measurements and applying deconvolution algorithms is the most promising approach to study cellular composition of organs and tissues, rather than improving single-cell genomics methods and applying them to profile molecular states of individual cells in each tissue type. However, I think the general consensus in the transcriptomics field and neuroscience is that in answering most questions about molecular changes during development and in disease, single-cell genomics have already become the gold standard and keep improving. This is evident from the sheer number of single-cell studies published over the recent years and the decline in the number of papers that utilize bulk gene expression measurements exclusively. There are many advantages of single-cell transcriptomics over bulk tissue gene expression combined with cellular deconvolution that are never mentioned by the authors: identifying novel cell types for which no markers exist, profiling tissues with unknown cell composition, differentiating between cell types that share a large proportion of signatures (e.g. subtypes of excitatory and inhibitory neuron subtypes) and identifying genes that change in specific cell types. Moreover, authors state current pitfalls of single-cell genomics but fail to mention that many of these issues are being currently addressed or have been addressed already:

“In practice, none of these approaches are feasible and cost effective for human transcriptome studies that require large sample sizes (hundreds to thousands of samples), such as eQTL studies or gene expression studies aiming to identify low magnitude changes in a disease group. Furthermore, sorting-based techniques require a priori knowledge of the cell type or sub-type of interest. Single-cell sequencing approaches, although able to identify cell types without prior selection, provide sparse data, with only a subset of genes detected per cell. In addition, the transcriptome complexity measured by scRNA-seq is limited, largely lacking measurements of non-coding RNAs and splicing isoforms. Therefore, our understanding of the genetic regulation of gene expression in human tissues, its variation during development and aging, and its dysregulation in disease, relies on transcriptome data derived from bulk tissue samples.”

The cost of single-cell sequencing keeps dropping thanks to development of multiplexing techniques (e.g. <https://doi.org/10.1038/s41592-019-0433-8>) and overall drop in sequencing costs. It is realistic to think we will be able to sequence thousands of samples in the next few years thanks to that. Gene detection has already improved dramatically, for instance using 10x Genomics Single Cell 3' v.3

compared to v.2 platform. It is simply not true that scRNA-seq cannot detect non-coding RNAs and spliced isoforms. Non-coding RNAs are detected as well as protein-coding genes; scRNA-seq will currently omit more genes expressed at low level compared to bulk RNA-seq regardless of their protein coding potential. It is true lncRNAs tend to be expressed at lower levels but you still detect many of them with scRNA-seq. Splicing analysis can be performed on the single-cell level (e.g. <https://www.sciencedirect.com/science/article/pii/S088875431930713X>).

Overall, I think authors send a wrong message by suggesting that generating new bulk RNA-seq data to study human brain development or its changes in disease is a viable option for most future studies of complex tissues. The authors do not need to spend too much time contrasting single-cell RNA-seq with bulk deconvolution but rather need to state that due to the amount of bulk tissue gene expression data that have already been generated, cellular deconvolution methods hold value to enable us to leverage new single-cell genomics datasets to deeper analyze existing bulk data.

2) Datasets used for the main analyses are not appropriate and not up-to-date. The Darmanis dataset is from 2015 and contains only a few hundreds of cells. This is the dataset used to perform the main analyses in the paper. I understand that these data were used because they are single-cell rather than single-nucleus but apart from containing very few cells, because these data are generated by isolating cells from adult human brain post resection, many cell types are partially or completely lost (especially subtypes of neurons). The resolution of this dataset is very low; for instance, there is no clear separation of excitatory and inhibitory neurons into subtypes, something we clearly see in more recent scRNA-seq studies in the mouse or snRNA-seq studies in the human.

For the sake of demonstrating the performance of deconvolution techniques, recent large-scale scRNA-seq data from the mouse can be used (<https://doi.org/10.1016/j.cell.2018.06.021>, <https://www.nature.com/articles/s41586-018-0654-5?spm=smc.content.content.1.154725120011217D5jli>).

If the goal is to develop and utilize a reference signature for the adult human brain, this will have to be done using snRNA-seq data since this is the type of data that is currently being generated and available for the adult human brain. Generating scRNA-seq data from the adult human brain is challenging and unreliable since many cells and cell types are damaged and lost during isolation procedure, and many brain regions are not surgically resected and thus will never be available for scRNA-seq. Utilizing snRNA-seq to deconvolute whole-tissue RNA-seq data might require adjustments to account for biases to actively transcribed/nuclear localized transcripts in snRNA-seq.

3) Biological assumptions and conclusions are oversimplified. Across the manuscript, the authors compare imputed vs actual cell type ratios for the five cell types: neurons, astrocytes, oligodendrocytes, microglia, and endothelia. The human brain contains tens to hundreds of cell types, including subtypes of excitatory and inhibitory interneurons and astrocytes. These cell types have vastly different morphologies, functions, connectivity patterns and are affected differentially in specific diseases. Lumping them into just five cell types is a gross oversimplification. Moreover, these subtypes can be readily distinguished using unbiased sc(sn)RNA-seq. My question is what is the value of being able to deconvolute bulk gene expression data to only five cell types when in reality say neurons are composed of tens of neuronal subtypes that can change very differently across development and in disease? The applications will be very limited to for instance assessing changes in overall neuronal numbers in neurodegenerative diseases. For the study's conclusions and observations to be widely applicable, the

authors need to attempt to deconvolve finer cell populations, such as subtypes of excitatory and inhibitory neurons.

Beyond estimating cellular composition and adjusting for it when performing differential gene expression analysis, can gene expression changes be attributed to specific cell types?

I don't think generating a cell signature reference by combining different datasets is a good idea. Cell types across different brain regions are known to be different, averaging neuronal gene expression signatures will lead to not accounting for the real regional neuronal diversity when performing deconvolution analyses.

Minor comments:

Introduction is rather technical in scope. Several sections can be moved to Methods, and the introduction needs to clearly state the current state of the field and issues the study is trying to address.

Reviewer #3 (Remarks to the Author):

In this manuscript, the authors compare the performance of different deconvolution methods and reference signatures for human brain expression data.

While the study presents some interesting results, I'm not really convinced by the messages the authors try to bring forward e.g. with limited novelty and not always well substantiated. Importantly, many other studies have already brought to our attention the importance of the reference signatures. [Vallania, F. et al. Leveraging heterogeneity across multiple datasets increases cell-mixture deconvolution accuracy and reduces biological and technical biases. *Nat. Commun.* 9, 4735 (2018).] The conclusion that this not holds true for pancreas cancer is not well substantiated and not convincing. Moreover, to prove the tissue specificity of deconvolution methods at least more than 2 tissue types should be included in the study to make convincing conclusions. In addition, the paper is lengthy and should be written more concise and with more structure.

Abstract

- Line 23: "The human brain is unique in its transcriptomic diversity...": this should be explained a bit more. It is not always very clear how and why brain deviates from other (complex) tissue types in the context of deconvolution questions.
- Line 29: In this manuscript, it is mentioned that 22 transcriptome deconvolution approaches are compared. However, it is misleading to formulate it in this way as it includes 3 methods for which 6 different categories of cell type signatures are used.

Results

- Line 195: "mixtures of 100 randomly sampled cells": What is the rationale to take 100 cells and not more? Is this enough to reconstruct a bulk sample?
- Figure 1: Were the same cells used for the mixtures as for the signature identification? Or were the cells split in a training (for reference) and test set (for mixtures)?

- Line 183: What was the rationale to include enrichment-based methods in this deconvolution benchmarking study? Proportions and scores are difficult to compare making this story a bit complex, not always allowing to perform the same statistics/analyses.
- Line 223: It is not clear why the authors didn't include mixtures with low abundant cell types, eg 0.05/0.95 and 0.95/0.05, which is a very relevant question in deconvolution studies (also given the authors' observation that errors were higher for cell types with lower abundance). In addition, including only 2 cell types in this mixture experiment makes it an oversimplified model.
- Line 266 (and line 454): I do not agree with the authors that their data suggest that for some tissues the signature genes are less influencing the deconvolution result than for other tissues. Cultured pancreas cells might be more stable and more similar to the freshly-isolated cells, which might explain the lack of effect of different signatures on deconvolution results. How different/similar are the top genes in the different signatures for the brain data (eg is LK very different from the rest) vs the pancreas data?
- Line 314: The position of the paragraph on the interplay between cell-type composition and DEGs: it might be more logical to position this paragraph at the start of the manuscript to generate the rationale for deconvolution?
- Line 369: "given that true cell type composition was not known": authors should include a real dataset where the cell composition is known (eg from cell sorting experiments on parallel samples) to make firm conclusions.
- In the manuscript it is not discussed what happens in case of a missing cell type in the reference datasets/signatures, which is an important issue in computational deconvolution of cell proportions?
- Line 461: reference to Figure X
- Line 368: The multibrain signature is only mentioned in the last paragraph: Why was it not used in the previous paragraphs?
- Figure 1: a white square is overlapping part of the figure

Reviewer #4 (Remarks to the Author):

In this manuscript, Sutton and Voineagu evaluate different methods for cell type deconvolution in brain tissue. They find that the choice of cell type reference (and specifically, the biology of the system used to derive the signature data) is the most impactful variable in the accuracy of the deconvolution methods; they also present some recommendations on the choice of reference and the class of method. The manuscript is sound and well-written, though some extra work would be required to convince me that the findings more generally applicable. I have listed my comments in more detail below.

1. What is the effect of variation in total RNA content across cell types on the performance evaluation? This will affect how the cell type proportions are interpreted, as the same amount of RNA in the bulk sample may be contributed by more or fewer cells depending on the total RNA in each cell type. In particular, differences in total RNA content are not reflected in the random sampling simulation as the averaging is done on per-cell RPKMs (where such differences have already been normalized), but in

reality, the mixture will have greater contributions from cell types with higher RNA content. The authors already touch on this issue on line 628, so I would have hoped for more consideration of this effect in the simulations as well.

2. What aspect of the MultiBrain expression profile provides the greatest benefit? Is it the fact that data is shared across three difference reference profiles, thus averaging out any study-specific effects? Or is it due to the extra quantile normalization step, which does not appear to have been applied to any of the other references?

3. The fact that the cellular composition "biases" the DE analysis is a fairly obvious application of Simpson's paradox. Of greater interest would be a discussion of how to account for differences in composition across bulk RNA-seq samples. For example, should we use the estimated compositions directly as covariates in a linear model? But if we do so, most methods for DE analyses use a log-link between the coefficients and the expected counts, and the compositions would only make theoretical sense under an identity link. Does this discrepancy affect performance? Or is it better to create spline basis matrices for the vector of compositions for each cell type, to account for any trend with respect to cell type composition? I am not aware of much work in this area, so this would be quite valuable to the community.

4. The authors might consider using a wider variety of single-cell datasets to generate their simulations. For example, the Zeisel and Tasic datasets have a diverse array of cell types, to name a few. Currently the simulations are generated from a single dataset, which makes it hard to determine the generality of the results. The authors might also consider augmenting their in vitro mixture data with in silico mixtures of pure profiles, e.g., using a leave-one-out scheme where one reference's profiles are used to construct the mixtures and the other references are used for deconvolution. This will provide more information about performance beyond the neurons and astrocytes used in the in vitro experiment.

5. A non-negligible amount of work has clearly gone into preparing the signature expression data from various reference datasets, in a form that is easily digestible by a wide variety of annotation methods. Exposing these "cleaned" datasets in a user-friendly manner would be of great value to the bioinformatics community - see, for example, the celldex Bioconductor package (<https://bioconductor.org/packages/devel/data/experiment/html/celldex.html>) which provides mostly immune references in an easily accessible format. The authors may consider making the datasets used in this manuscript available in a similar manner, e.g., by contributing to celldex or by creating their own Bioconductor package. I'm sure that people in Jean Yang's group over in USyd would be happy to advise.

6. I would like to see the evaluation repeated for more tissues. The authors have done so for pancreas, but I would also be curious about other complex systems (e.g., blood, retina, mammary glands). Is the brain particularly special in its performance characteristics? This is difficult to answer by having just one other tissue to compare it to.

7. I don't see any GEO submission for the sequencing data.

We thank the reviewers for the very constructive comments and prompting us to do some important analyses. We have carried out an extensive revision and we believe the manuscript is substantially improved.

In summary, the revised manuscript contains the following new components:

- A **new dataset**, which we generated to investigate the effect of using snRNA-seq to deconvolve bulk RNA-seq from whole tissue samples (in response to Rev2). We generated snRNA-seq as well as paired bulk RNA-seq from whole tissue and nuclear extracts, from the same brain samples. These data allowed us to identify compartment-specific genes and to assess their effect on deconvolution.
- **Major new analyses**
 - All benchmarking analyses have been re-done on two additional larger datasets (in response to Rev 2 and 3).
 - A single-cell based algorithm has been added to the benchmarked methods.
 - Benchmarking has been extended to another non-brain tissue, now including both pancreas and heart (in response to Rev 3 and 4).
 - We investigated various statistical approaches for correcting cell-type composition differences in differential expression analyses (in response to Rev4).
- **New resources**
 - We developed an R package including all the brain cell-type signatures (as suggested by Rev 4).
 - We also developed a web tool which implements deconvolution with the top performing algorithms and all the different cell-type signatures and provides goodness-of-fit for the input data.

The reviewers' comments are addressed point-by-point below. The corresponding changes in text are highlighted in red in the revised manuscript, and the most relevant text updates are included in this letter in italics.

Reviewer #2

Major comments:

1) The introduction is rather biased toward listing pros of bulk gene expression deconvolution methods and cons of single-cell transcriptomic methods.... The authors do not need to spend too much time contrasting single-cell RNA-seq with bulk deconvolution but rather need to state that due to the amount of bulk tissue gene expression data that have already been generated, cellular deconvolution methods hold value to enable us to leverage new single-cell genomics datasets to deeper analyze existing bulk data.

As suggested, we have replaced the text focusing on technical limitations of scRNA-seq, with the following paragraph (Page 3): *In silico deconvolution offers the opportunity to leverage scRNA-seq data to obtain deeper insights into bulk tissue transcriptomes generated through large-scale studies such as GTEX¹⁴, PsychEncode¹⁵, the Common Mind Consortium¹⁶, and BrainSpan¹⁷.*

2) Datasets used for the main analyses are not appropriate and not up-to-date. The Darmanis dataset is from 2015 and contains only a few hundreds of cells. This is the dataset used to perform the main analyses in the paper. Recent large-scale scRNA-seq data from the mouse can be used (<https://doi.org/10.1016/j.cell.2018.06.021>, <https://www.nature.com/articles/s41586-018-0654-5?spm=smc.content.content.1.154725120011217D5jli>).

To address this comment, we have now carried out all the benchmarking analyses on simulations based on two additional large single-cell datasets from human brain:

- 10X Chromium data from Velmeshev *et al.*, Science, 2019 (VL). This dataset contains over 100,000 nuclei.
- Smart-seq2 data from the Human Cell Atlas (CA) (Hodge *et al.*, Nature 2019). This dataset contains over 15,000 nuclei with deeper coverage than the 10X Chromium data.

We selected these datasets as they are generated from the human brain, the focus of our paper, and the number of nuclei is on the same order of magnitude as the mouse datasets suggested (Tasic *et al.* ~23,000; Ziesel *et al.* ~500,000). We included the data from Tasic *et al.* as a mouse cell-type signature data (TS).

We now present the simulation data from Velmeshev *et al.* as the primary results (Page 4, Fig.1) and the simulations based using the CA and Darmanis datasets as replication in supplementary materials. These new data confirm our initial observations that the cell-type signature data has a stronger impact on deconvolution than the choice of algorithm. In addition, these datasets allowed us to investigate the deconvolution of finer cell-type populations as

suggested (see point #4), and were also included as signatures for the deconvolution of GTEx and PsychEncode data (Fig.5).

3) If the goal is to develop and utilize a reference signature for the adult human brain, this will have to be done using snRNA-seq data since this is the type of data that is currently being generated and available for the adult human brain. Utilizing snRNA-seq to deconvolute whole-tissue RNA-seq data might require adjustments to account for biases to actively transcribed/nuclear localized transcripts in snRNA-seq.

We thank the reviewer for this important question. To investigate the effect of using snRNA-seq data on deconvolving bulk tissue data we generated paired bulk RNA-seq and nuclear RNA-seq from the same samples. Five frozen brain tissue samples were pulverised and split to extract RNA from either the nuclear fraction or whole tissue, followed by sequencing; the nuclear-specific transcript *MALAT1* was 21-fold enriched in the nuclear fractions (Methods; Supplementary Table 8). We also generated snRNA-seq from the same brain tissue samples.

We used these new data to identify **compartment-specific genes** i.e. nuclear-localised or nuclear-depleted transcripts, which as the reviewer suggests, may impact deconvolution of tissue data using snRNA-seq. Overall, we find that removing compartment-specific genes from the cell-type signature data improves the deconvolution performance when using snRNA-seq based signatures. We include this new analysis as a sub-section in Results, Page 6.

- *The effect of compartment specific genes on deconvolution accuracy*

Since most single-cell data from the adult human brain are generated using single-nucleus RNA-seq, while bulk RNA-seq is based on total RNA, we next investigated whether compartment-specific genes (i.e. those either enriched or depleted from the nucleus) influence the outcome of deconvolution. For this purpose, we generated paired bulk RNA-seq and nuclear RNA-seq from five frozen brain tissue samples (Methods), as well as snRNA-seq from the same brain samples. We identified compartment-specific genes as those differentially-expressed between the nuclear and total bulk RNA-seq ($FDR < 0.05$, $|FC| > 1.3$; Supplementary Table 1). We then carried out several deconvolution analyses with and without filtering-out the compartment-specific genes.

Firstly, we deconvolved the twenty-one bulk RNA-seq samples from sorted brain cells³⁸, where true cell-type composition is known (i.e. each sample is expected to be a nearly-pure cell-type, with some experimental variability of the sorting efficiency). We deconvolved these data with either the identical cell-type signature (derived from the sorted dataset; IP), an scRNA-seq signature (DM), and four snRNA-seq signatures (VL, CA, NG, and LK); Supplementary Table 2. When using the IP signature, the immuno-panned cell-type was estimated as $> 80\%$ abundant in all samples. Thus we assessed the proportion of samples in which the sorted cell-type was correctly identified (i.e. estimated proportion $> 80\%$) using the scRNA-seq and snRNA-seq signatures, with or without filtering out compartment-specific genes. We found that the snRNA-seq signatures performed well even prior to filtering out compartment-specific genes, correctly identifying the sorted cell-type in an average of 86% of samples (71%-95%). As expected, the single-cell-based signature (Supplementary Fig.25) performed somewhat better, identifying the sorted cell-type in an average of 90% of samples. Removing compartment-specific genes further improved the outcome for snRNA-seq signatures: the sorted cell-type was correctly identified in an average of 88% of samples (86%-95%); Supplementary Fig.25, eliminating the difference between the scRNA-seq and snRNA-seq signatures.

To increase the complexity of the deconvolution task, we asked how accurately the five whole-tissue samples were deconvolved when using snRNA-seq data from the same individuals, as compared to a whole-cell based signature (Supplementary Fig.25). In this case, if compartment-specific genes were not removed from the cell-type signature, the correlation between cell-type proportions estimated using the snRNA-seq signature and the whole-cell signature was modest ($r=0.27$). However, the correlation improved substantially by filtering out compartment-specific genes ($r=0.98$) suggesting that this filtering approach should be considered when using snRNA-seq-based cell-type signatures.

Supplementary Figure 25. The role of compartment-specific genes when using snRNA-seq signatures. A and B: Estimated proportions for pure samples of immuno-panned cell-types, using whole-cell and snRNA-seq signatures. (A) all genes were included in the signature. (B) compartment-specific genes were filtered-out. *Thick horizontal line:* mean. *Dotted vertical line:* separates whole-cell from snRNA-seq signatures. *Neu:* neurons. *Ast:* astrocytes. *Oli:* oligodendrocytes. *Mic:* microglia. **C.** Scatterplot of estimated proportions for bulk brain samples using the IP bulk RNA-seq signature (x-axis) or the snRNA-seq signature derived from the same individual (y-axis). *Left:* all genes were included in the signature; *right:* compartment-specific genes were filtered-out. *Individual:* NICHD brain bank id number. Cell-type proportions were estimated using CIBERSORT.

4) For the study's conclusions and observations to be widely applicable, the authors need to attempt to deconvolve finer cell populations, such as subtypes of excitatory and inhibitory neurons.

This is another important question, and we thank the reviewer for prompting us to further investigate it.

We have assessed the impact of deconvolving finer cell populations on the deconvolution outcome using the VL- and CA- based simulations. Overall, we found that accurate deconvolution can be achieved for neuronal sub-types, if they are not lowly abundant (< 2%) or highly collinear with other cell-types (correlation of gene expression levels between cell-types $\rho > 0.95$). The following section is now included in Results, page 5.

- **Deconvolution of cellular subtypes**

We next explored how including cellular sub-types affected deconvolution accuracy for brain data. First, we used broad neuronal sub-types, i.e. excitatory and inhibitory neurons (Fig.1B middle, Supplementary Fig.3,9), and found that deconvolution accuracy was high ($r > 0.8$ for all algorithms), with *cib* performing best ($r = 0.94$ and 0.95 for excitatory and inhibitory, respectively). The accuracy for the other cell-types was largely unaffected by neurons being sub-classified (Fig.1B, middle, Supplementary Fig.3,9). This result was replicated in the CA-based simulations (Supplementary Fig.1,5,10).

When including all cell sub-types detected in the VL dataset (11 neuronal and 2 astrocyte sub-types), deconvolution with *cib* remained accurate ($r > 0.8$) for most cell populations (Fig.1B, right, Supplementary Fig.3,11,12). However, two main factors led to a reduction in accuracy for certain cell-types: low abundance of the cell-type (< 2%) and high collinearity (gene expression correlation with another cell type $\rho > 0.95$); Supplementary Fig.13,14. This observation was replicated in the CA dataset with most cell sub-types being accurately deconvolved ($r > 0.8$; Supplementary Fig.1,5,15,16) and collinearity being the main factor that led to reduced accuracy (Supplementary Fig.17,18).

Figure 1B. Barplots of Pearson correlation coefficients (r) between true and estimated cell-type proportions in 100 *in silico* mixtures. *Left*: cells are grouped by major cell-types; *middle*: excitatory and inhibitory neuron subtypes are included in the signature; *right*: all cell-subtype labels are used in the signature. *Dotted line*: $r = 0.8$.

5) Beyond estimating cellular composition and adjusting for it when performing differential gene expression analysis, can gene expression changes be attributed to specific cell types?

We thank the reviewer for this suggestion, which we address in detail in the revised manuscript. CIBERSORTx, the newest version of CIBERSORT, provides gene expression estimates in each cell-type in addition to cell-type proportion estimates. These cell-type specific expression values can be used to carry out cell-type specific differential expression analyses. We generated simulated data to assess whether this approach correctly assigns differentially expressed genes to the cell-type where they are perturbed. We found that using cell-type specific gene expression for DE analyses is effective at detecting DE genes in the right cell-type when the gene expression and composition changes are not confounded, but this comes at the expense of a moderate increase in false-positives. These new data are presented in Results, Pages 8-9:

We also investigated whether the cell-type where differential expression occurs can be uncovered through deconvolution. To this end, we used CIBERSORTx⁴⁸, which takes a bulk mixture and estimates gene expression values in each cell-type present in the signature data; these cell-type-specific expression values can then be used to carry out cell-type specific DE analyses. We thus simulated data with 1.5-fold change in expression specific to a given cell-type, with or without superimposed differences in cell-type composition between the two groups (Supplementary Fig.32), and tested whether genes were identified as DE in the correct cell-type. As above, the 1.5-fold expression difference was simulated for 200 genes, of which 100 are markers of the perturbed cell-type and 100 are non-marker genes.

The expression difference was first simulated in excitatory neurons. In the absence of confounding cell-type composition differences between the two groups, more than 95% of the perturbed excitatory marker genes were detected as DE in the correct cell-type (excitatory neurons), while for the non-marker genes ~45% were detected as DE in excitatory neurons and another ~30% were incorrectly detected as DE in inhibitory neurons; Fig. 4E. The false-positive rate (i.e. the fraction of non-perturbed genes detected as DE) was 0% (Supplementary Fig.32).

When a composition difference was superimposed (~10% increase in excitatory neurons; Methods, Supplementary Fig.32), the true-positive rate was unchanged except for upregulated marker genes, where it was reduced, likely due to the fact that the composition change and the expression change were confounded (both variables were higher in group B vs. group A). The false-positive rate was less than 12% in all cell-types, thus drastically reduced relative to no correction for cell-type composition (32%), but higher than when correcting for composition differences in a standard linear model (0%). Similar results were observed when the gene expression change was modelled in inhibitory neurons (Supplementary Fig.32).

Overall, these results suggest that using cell-type specific gene expression for DE analyses is effective at detecting DE genes in the right cell-type when the gene expression and composition changes are not confounded, but this comes at the expense of a moderate increase in false-positives.

6) I don't think generating a cell signature reference by combining different datasets is a good idea. Cell types across different brain regions are known to be different, averaging neuronal gene expression signatures will lead to not accounting for the real regional neuronal diversity when performing deconvolution analyses.

We apologize that the description of the MultiBrain signature was not clear – we only averaged data from different datasets of the same brain region (cerebral cortex). This approach has previously shown to be effective for blood data (PMID: 30413720), by averaging out inter-individual and technical differences between datasets.

The MultiBrain signature (which now also includes cortex data from VL and CA) performs better than other signatures on the Parikshak *et al.* cortex dataset, and among the top 2 on the cortex GTEx data (Fig.5). We now clarify in the methods section (Page 15) that the MultiBrain signature includes data from cortex only, and as suggested by Rev 4, we generated an R package which includes all the individual signatures benchmarked here and also developed a web tool implementing deconvolution with the best performing algorithm and all the cell-type signatures, allowing users to calculate goodness-of-fit and determine the appropriate cell-type signature for their dataset of interest.

Minor comments: Introduction is rather technical in scope. Several sections can be moved to Methods, and the introduction needs to clearly state the current state of the field and issues the study is trying to address. As suggested, we have moved the technical description of deconvolution methods to the Methods section.

Reviewer #3

Many other studies have already brought to our attention the importance of the reference signatures. [Vallania, F. et al. Leveraging heterogeneity across multiple datasets increases cell-mixture deconvolution accuracy and reduces biological and technical biases. *Nat. Commun.* 9, 4735 (2018).]

While this is true and we already cite those studies (ref#61 in the original manuscript), the properties of cell-type signature that are important for deconvolution vary across tissues. Vallania *et al.* showed that for blood-derived signatures the microarray platform was the main factor driving differences between cell-type signature datasets, while for solid tumors, biological factors, in particular the tumor environment, are the main factors that drive the performance of cell-type signature data (Schelker, M. *et al. Nat. Commun.* 8, 2032 (2017).

Here we carry out the first study that identifies the properties of the cell-type signature important for deconvolution of Brain transcriptomes.

...to prove the tissue specificity of deconvolution methods at least more than 2 tissue types should be included in the study to make convincing conclusions.

We agree with this point and have extended the analyses as suggested. Please see response to point #7 below.

Abstract

1. Line 23: “The human brain is unique in its transcriptomic diversity...”: this should be explained a bit more. It is not always very clear how and why brain deviates from other (complex) tissue types in the context of deconvolution questions.

We have added the following brief explanation, in the limit afforded by the Abstract section:

The human brain is unique in its transcriptomic diversity, expressing the highest diversity of alternative splicing isoforms and non-coding RNAs. It comprises a complex mixture of cell-types including transcriptionally similar sub-types of neurons, which undergo gene expression changes in response to neuronal activity.

2. Line 29: In this manuscript, it is mentioned that 22 transcriptome deconvolution approaches are compared. However, it is misleading to formulate it in this way as it includes 3 methods for which 6 different categories of cell type signatures are used.

As suggested, we updated the abstract to clarify the difference between the number of methods and the number of signatures applied (note that we have added one new partial deconvolution method – MuSiC – and 3 new signatures – Velmeshev *et al.*, Nagy *et al.* and Tasic *et al.* – in the revised manuscript):

“We evaluate 8 transcriptome deconvolution approaches, covering all main classes: 4 partial deconvolution methods, each applied with 9 different cell-type signatures, 2 enrichment methods and 2 complete deconvolution methods”

Results

3. Line 195: “mixtures of 100 randomly sampled cells”: What is the rationale to take 100 cells and not more? Is this enough to reconstruct a bulk sample? This choice was determined by the number of cells in the Darmanis *et al.* dataset used for simulations (n=297). We have now added two more datasets with larger numbers of cells (see Rev2 point #2) and for these datasets we have generated the mixtures using 500 cells. We found that the results are overall consistent between datasets (Results, page 4), and thus robust to the choice of number of cells used for sampling. In addition, we added a supplementary figure demonstrating that the gene expression distribution in our simulated mixtures is similar with the distribution of bulk RNA-seq data (Supplementary Figure 41).

Supplementary Figure 41. Distribution of gene expression values in real and simulated brain mixtures.

Brain: bulk brain RNA-seq from Parikshak *et al.* (2016). *Simulations*: *in silico* mixtures simulated from the corresponding dataset. *DM*: scRNA-seq from Darmanis *et al.*. *CA*: snRNA-seq data from the Human Cell Atlas. *VL*: snRNA-seq data from Velmeshev *et al.* (2019). Simulations contained 500 cells/mixtures for CA and VL and 100 cells/mixture for DM. *Nuclei, Cells*: single-nuclei and single-cells from the corresponding dataset. Ten samples were randomly for each plot to minimise overplotting. Note: DM simulation is in units of RPKM rather than CPM.

4. Figure 1: Were the same cells used for the mixtures as for the signature identification? Or were the cells split in a training (for reference) and test set (for mixtures)?

Yes, in the original manuscript using data from Darmanis *et al.*, cells were not split for mixtures and signatures due to the low numbers of cells in the dataset. In the revised manuscript, for the two new large datasets used for simulations, we followed the reviewer's suggestion and randomly split the cells into two equally-sized pools, with one pool used for mixtures, and the other for signatures (Fig.1).

5. Line 183: What was the rationale to include enrichment-based methods in this deconvolution benchmarking study? Proportions and scores are difficult to compare making this story a bit complex, not always allowing to perform the same statistics/analyses.

We agree that enrichment-based methods are difficult to evaluate. However, large scale projects such as GTEx have carried out deconvolution using enrichment methods (xCell) to derive cell-type specific eQTLs (PMID: 32913098), and thus we thought it was important to benchmark the performance of these methods.

6. Line 223: It is not clear why the authors didn't include mixtures with low abundant cell types, e.g. 0.05/0.95 and 0.95/0.05, which is a very relevant question in deconvolution studies (also given the authors' observation that errors were higher for cell types with lower abundance). In addition, including only 2 cell types in this mixture experiment makes it an oversimplified model.

We thank the reviewer for pointing this out, it is indeed an important aspect. Using the new Velmeshev-derived simulations, we now include several cell-types and sub-types with abundances < 5% (5 excitatory neuron sub-types, 3 inhibitory neuron sub-types, 2 astrocyte sub-types, endothelia, microglia, OPCs, and oligodendrocytes). We found that deconvolution accuracy was reduced when cell-type abundance fell below 2% (Results, Page 4).

7. Line 266 (and line 454): I do not agree with the authors that their data suggest that for some tissues the signature genes are less influencing the deconvolution result than for other tissues. Cultured pancreas cells might be more stable and more similar to the freshly-isolated cells, which might explain the lack of effect of different signatures on deconvolution results. How different/similar are the top genes in the different signatures for the brain data (eg is LK very different from the rest) vs the pancreas data?

We agree that our initial evaluation of the effect of signatures across tissues was insufficient, as it was done for a single non-brain tissue (pancreas), where the simulations contained only two cell-types. To address this question more comprehensively, we now investigate deconvolution accuracy of GTEx data for pancreas, heart left ventricle and arterial appendage, for which we could obtain cell-type signatures derived from either tissue or cultured cells – as *in-vitro* culturing is the main biological factor we uncovered here. Overall, we found that using signature data from cultured cells strongly affected the deconvolution accuracy for heart left ventricle and arterial appendage but mildly affected accuracy for pancreas data. These results indicate that – as expected – biological factors such as *in-vitro* culturing vary in their effect on gene expression across tissues. This is reflected in the suggested comparison of similarity of gene expression signatures (Supplementary Figure 38 - correlation coefficients, Supplementary Figure 39 - overlap of the top 100 marker genes). We present these data in the Results section, page 9-10:

While the important role of signature data has been previously reported³⁹, here we uncovered the effect of novel biological factors that affect deconvolution accuracy, in particular in-vitro culturing and brain region. Since in-vitro culturing may be relevant to other tissues as well, we investigated whether using cultured-cell-derived or tissue-derived signature data influences the accuracy of deconvolution for two additional tissues in GTEx: pancreas and heart. We found that deconvolution accuracy for left heart ventricle and arterial appendage samples was significantly reduced when using signature data from cultured cells, while the deconvolution accuracy for pancreas

data was mildly reduced (Supplementary Fig.37). These data suggest that the influence of biological factors on cell-type signature data vary across tissues, indicating that tissue-specific benchmarking of deconvolution approaches is warranted.

Supplementary Figure 37. Violin plots of goodness-of-fit in two non-brain tissues in the GTEx dataset.

A. Pancreas samples. **B.** Heart left ventricle samples. **C.** Heart atrial appendage samples. *Fresh*: signature derived from freshly-processed human tissue. *Cultured*: signature derived from cultured cells. The bottom, middle, and top of the white boxes mark the first, second, and third quantiles, respectively.

We also amended the discussion section (Page 10-11), removing the statement that cell-type signature does not affect the deconvolution for pancreas data, and the text now reads:

“In vitro culturing also affected the performance of deconvolution for other tissues (heart and pancreas), but to different extents (Supplementary Fig.37), highlighting the importance of tissue-specific benchmarking.”

8. Line 314: The position of the paragraph on the interplay between cell-type composition and DEGs: it might be more logical to position this paragraph at the start of the manuscript to generate the rationale for deconvolution?

As suggested, we in the revised manuscript we include the motivating paragraph in introduction (Page 3) and start the results section with a statement of the analyses carried out.

9. Line 369: “given that true cell type composition was not known”: authors should include a real dataset where the cell composition is known (eg from cell sorting experiments on parallel samples) to make firm conclusions.

We thank the reviewer for this suggestion. We now include deconvolution of a dataset of bulk RNA-seq generated by cell-sorting with immuno-panning. The results are consistent with the observations made on simulated data and on RNA mixtures and are included in the Results section page 4, Supplementary Fig.8.

10. In the manuscript it is not discussed what happens in case of a missing cell type in the reference datasets/signatures, which is an important issue in computational deconvolution of cell proportions?

We thank the reviewer for this important suggestion. We have now added a sub-section addressing this question (Results, page 5).

We also explored how deconvolution was affected when a cell-type was missing from the signature. We removed one cell-type at a time from the VL-derived mixtures, and found that when an abundant cell-type was missing (Neurons, 87.4%), the deconvolution accuracy was substantially reduced (mean r was reduced from 0.85 to 0.41, and normalised mean absolute error increased from 0.33 to 10.3). However, when lowly-abundant cell-types were missing from the signature, the effect on deconvolution was minimal (Supplementary Figure 19). We then tested the effect of removing a sub-type of neurons, excitatory or inhibitory neurons, which are highly correlated in expression ($\rho=0.92$). Deconvolution accuracy was reduced to a lesser extent than when all neurons were missing from the signature: r was reduced from 0.87 to 0.71 when excitatory neurons were missing, and from 0.86 to 0.76 when inhibitory neurons were missing (Supplementary Figure 19).

Supplementary Figure 19. Effect of removing cell-types or cell-subtypes from the signature matrix.

For each cell type, its mean abundance in the mixtures is shown in brackets, and scatterplots display the deconvolution accuracy when all cell types are present in the signature (x-axis) vs. when the cell type is absent from signature (y-axis). Accuracy is measured as either r correlation coefficient (left panel) or normalised mean absolute error (right panel). Calculations of mean r and mean NMAE for the x-axis label do not include the absent cell-type, and thus differ across plots. *Dotted red line: $y = x$.*

11. Line 461: reference to Figure X – thank you, fixed.

12. Line 368: The multibrain signature is only mentioned in the last paragraph: Why was it not used in the previous paragraphs?

We thought that the MultiBrain signature was particularly relevant for deconvolving large-scale datasets like GTEx and PsychENCODE data on ASD, where averaging out inter-individual variation was an important factor, while in our simulations, such variation is minimal.

13. Figure 1: a white square is overlapping part of the figure— thank you, fixed.

Reviewer #4

1. What is the effect of variation in total RNA content across cell types on the performance evaluation? The authors already touch on this issue on line 628, so I would have hoped for more consideration of this effect in the simulations as well.

We thank the reviewer for pointing us to further discuss this point. As previously demonstrated (*e.g.* PMID 31101809), due to the difference in RNA content across cell-types, deconvolution of RNA-seq data estimates RNA proportions rather than cell proportions. Since RNA-seq sequences a mixture of RNA molecules (primarily protein coding, after poly-A selection or ribo-depletion) we believe the goal of transcriptome deconvolution is to estimate the proportion of the sequenced RNA molecules coming from a given cell-type (pRNA), rather than the proportion of cells, and it is pRNA (not pCt) that is appropriate for reconstruction of gene expression data. This latter point is important to highlight, because – perhaps as a notion inherited from deconvolution of DNA methylation data – it is often assumed that one should accurately estimate cell-type proportions (pCt) from transcriptome data.

As suggested, in the revised manuscript we explore the notions above in our simulated data (Pages 23-24):

To test this hypothesis, we deconvolved pseudo-bulk data from the Velmeshev et al. snRNA-seq dataset (10 A-priori, pRNA (not pCt) should be relevant for reconstruction of gene expression data, and thus useful as a co-variate in differential expression analyses. To test this hypothesis, we deconvolved pseudo-bulk data from the Velmeshev et al. snRNA-seq dataset (10 individuals), where we know both pCt and pRNA (calculated as the proportion of RNA-seq reads from each cell-type), and found that deconvolution estimates perfectly correlate with pRNA but less so with pCt (Supplementary Figure 43), consistent with previous data³³. Note that pRNA and pCt are themselves correlated in this dataset ($r=0.86$). We then assessed goodness-of-fit for these pseudo-bulk data using either pRNA or pCt to reconstruct gene expression. We found that goodness-of-fit was always higher when using pRNA (Supplementary Figure 43). These data demonstrate that pRNA, the output of transcriptome deconvolution, is the appropriate measure to use for re-constructing gene expression data and thus as a co-variate in DE analyses.

Supplementary Figure 43.

The relationship between deconvolution estimates, RNA proportions (pRNA) and cell-type proportion (pCt).

A. Boxplots of RNA content per cell, reflected in the number of unique molecular identifiers (UMIs) per-cell across cell types in the VL dataset. *Ast*: astrocytes. *End*: Endothelia. *Exc*: Excitatory Neurons. *Inh*: Inhibitory Neurons. *Mic*: Microglia. *Oli*: Oligodendrocytes. *OPC*: Oligodendrocyte Precursor Cells. **B.** Scatterplot of pCt vs. pRNA in pseudo-bulk samples from 10 individuals. **C.** Scatterplot of Estimated proportion (y-axis) versus true pCt (left) or pRNA (right). **D.** Scatterplot of goodness-of-fit when reconstructing gene expression using pCt or pRNA. *Dotted black lines: $y=x$*

2. What aspect of the MultiBrain expression profile provides the greatest benefit? Is it the fact that data is shared across three difference reference profiles, thus averaging out any study-specific effects? Or is it due to the extra quantile normalization step, which does not appear to have been applied to any of the other references?

We believe the former is the case, *i.e.* it averages out study-specific biological and technical differences, including individual variation. This has been previously proposed (PMID: 30413720) and holds in our results.

3. The fact that the cellular composition "biases" the DE analysis is a fairly obvious application of Simpson's paradox. Of greater interest would be a discussion of how to account for differences in composition across bulk RNA-seq samples. For example, should we use the estimated compositions directly as covariates in a linear model? But if we do so, most methods for DE analyses use a log-link between the coefficients and the expected counts, and the compositions would only make theoretical sense under an identity link. Does this discrepancy affect performance? Or is it better to create spline basis matrices for the vector of compositions for each cell type, to account for any trend with respect to cell type composition? I am not aware of much work in this area, so this would be quite valuable to the community.

We thank the reviewer for this suggested. We now investigated, together with a statistician (J. Gagnon-Bartch; U.Mich), the effect of correcting for cellular composition using either directly the estimated compositions as covariates, quadratic regression or a spline matrix. We found that using the composition estimates as covariates effectively corrected for cell-type composition confounds, with the magnitude of the confound that could be corrected depending on the magnitude of gene expression fold-changes (Results, pages 7-8). Using either quadratic regression or splines did not appear to have any additional benefit (Fig.4).

We next investigated the more challenging case where there are true differences in gene expression between the two groups, in addition to differences in cell-type composition. To this end, we simulated data with differences in cell-type composition as above, while also introducing gene expression changes in a set of 200 genes, of which 100 are markers of excitatory neurons and 100 are non-marker genes (Methods). Several sets of simulations were generated with a mean expression difference between groups of 1.1-, 1.3-, 1.5- or 2-fold.

To quantify how effectively cellular composition was corrected for, we calculated discriminatory power as the fraction of the 200 perturbed genes that were in the top 200 most significant DE genes. This measure rewards true-positives while penalising false-positives. We found that without correction, the discriminatory power decreased with the magnitude of cell-type composition difference between the two groups (Fig.4C; uncorrected). Correction for cell-type composition was effective at restoring discriminatory power for gene expression differences of 1.5 fold when composition differences were up to 12.5% (Fig.4D; corrected). As expected, for expression differences of lower magnitude (1.1 and 1.3) the effective correction range was narrower (6.3% and 6.9% respectively), while for stronger expression differences (2-fold) the effective correction range was wider (25.7%); Fig.4D, Supplementary Fig.31. All correction approaches performed similarly in this analysis, with the exception of spline regression which we found to be less effective (Fig.4C,D, Supplementary Fig.31).

Figure 4. C&D C. Scatterplot of the discriminatory power, *i.e.* fraction of the 200 perturbed genes in the top 200 most significantly differentially expressed genes (y-axis) versus simulated difference in excitatory neuron proportion between sample groups (x-axis) for simulated 1.5-fold expression difference. *Coloured lines:* local regression line. *Dotted line:* expected discriminatory power, *i.e.* 0.95 times the discriminatory power in the absence of cell-type composition differences between groups. **D.** Model robustness to cell-type composition differences across a range of fold-changes, quantified as the smallest composition change where discriminatory power fell below its expected value.

4. The authors might consider using a wider variety of single-cell datasets to generate their simulations. For example, **the Zeisel and Tasic** datasets have a diverse array of cell types, to name a few. Currently the simulations are generated from a single dataset, which makes it hard to determine the generality of the results.

As suggested, we added simulations based on two new larger datasets. Please see response to Rev.2 point 2, who had the same comment.

The authors might also consider augmenting their in vitro mixture data with in silico mixtures of pure profiles, e.g., using a leave-one-out scheme where one reference's profiles are used to construct the mixtures and the other references are used for deconvolution. This will provide more information about performance beyond the neurons and astrocytes used in the in vitro experiment.

If we understand correctly the comment, the reviewer suggests that (a) we use one sc/sn RNA-seq dataset to generate mixtures, and another dataset as reference for deconvolution or (b) we split cells from a given dataset and use a subset for mixtures and the rest for signature data. We have done both in the revised manuscript, on the larger datasets (VL and CA). Each dataset was split into two subsets to create mixtures and cell-types signature data. The mixtures were then deconvolved using either the “matched” signature data or signatures generated from several independent datasets (Figs.1-2, Supp Figs.20-24).

5. A non-negligible amount of work has clearly gone into preparing the signature expression data from various reference datasets, in a form that is easily digestible by a wide variety of annotation methods. Exposing these

"cleaned" datasets in a user-friendly manner would be of great value to the bioinformatics community - see, for example, the `celldex` Bioconductor package (<https://bioconductor.org/packages/devel/data/experiment/html/celldex.html>) which provides mostly immune references in an easily accessible format. The authors may consider making the datasets used in this manuscript available in a similar manner, e.g., by contributing to `celldex` or by creating their own Bioconductor package. I'm sure that people in Jean Yang's group over in USyd would be happy to advise.

We thank the reviewer for this suggestion. We created an R package that includes all the signatures from this benchmarking study (<https://github.com/Voineagulab/brainyR>), currently being submitted to Bioconductor. To further facilitate the use of these signatures we developed a shiny app which implements the best performing algorithms and allows the user to test the goodness of fit for their data using any of the signatures in the R package (<https://voineagulab.shinyapps.io/BrainDeconvShiny/>).

6. I would like to see the evaluation repeated for more tissues. The authors have done so for pancreas, but I would also be curious about other complex systems (e.g., blood, retina, mammary glands). Is the brain particularly special in its performance characteristics? This is difficult to answer by having just one other tissue to compare it to.

We agree that the comparison with a single other tissue limited the interpretation. We have now added assessment of deconvolution accuracy of GTEx data from heart left ventricle and atrial appendage, in addition to pancreas. We selected these tissues as we were able to obtain cell-type signatures derived from tissue as well as cultured cells. *In-vitro* culturing was the main biological factor we identified to be important for brain, and thus were interested to test whether the observation can be extended to other tissues. The results are described in Results pages 9-10, and included in this letter in response to Rev.4 point 7 who raised a similar question.

We would like to note that an analysis of non-brain tissues as comprehensive as the one we carried out for brain is beyond the scope of this study, as it would require biological expertise in each specific system.

7. I don't see any GEO submission for the sequencing data. Thank you for noticing this omission, the GEO ID is now included (Page 24).

REVIEWERS' COMMENTS

Reviewer #2 (Remarks to the Author):

In the resubmission of the manuscript, Sutton et al present a revised and much improved version of the paper, addressing most of my concerns and presenting results of analyzing additional more recent datasets. I thank the authors for the great body of work they added and for their meticulous revisions. I have last, but important comment related to a concern I raised during the first review. In my opinion, the ability to get accurate cell type-specific estimates from bulk data would be the strongest showcase of bulk deconvolution methods. Even though the authors added cell type-specific differential expression analysis of bulk data using CIBERSORTx in Supplementary Fig. 32, these important results are buried deep in the supplementary figures. These data should be added to one of the main figures. Moreover, I think real, rather than simulated data should be used for this analysis. Authors could try generating pseudo-bulk estimates by pulling snRNA-seq profiles by sample and applying CIBERSORTx to pseudobulk. Additionally, CIBERSORTx could be applied to the autism and control bulk data from the Parikshak et al. This would greatly strengthen the case of applying bulk deconvolution methods in the era of single-cell genomics.

Reviewer #3 (Remarks to the Author):

In general, the authors have answered the questions from the reviewers in a good way, however, I'm still not convinced about the novelty of the study compared to existing publications. The added value of the focus only on brain tissue doesn't convince me a lot.

Comment 1: Some words on the complexity of the brains' transcriptome should also be given in the introduction, to explain why this manuscript focuses on brain tissue specifically.

Comment 3: Based on the new supplementary figure I do not agree that the mixtures of 100 cells from the Darmanis paper have a similar distribution compared to the bulk and other mixtures (lower log-cpm values).

Introduction (or discussion) misses part on previous benchmarking studies on computational deconvolution tools.

Both in the main figure legends and the supplementary figure legends I sometimes miss annotation info (eg. what is CA, VL, TS, LK, NG, ...) and there are some inconsistencies in the use of abbreviations for example for Music (sometimes music and sometimes mus: main vs supplementary figures).

Line 158: I miss a more objective way to indicate which method performs best.

Line 161: why is the deconvolution performance comparison not performed using all methods (but only on 3 methods)? In this analysis, I also miss a mixture with low(er) abundant cell fraction, eg 10% vs 90% or 5% vs 95%...

Line 163: "In both cases, the deconvolution accuracy was very high when the signature was derived from the same source as the mixtures": what is meant by this sentence? Not clear to me. From a reply to one of my comments (comment 4), I thought that in this version analysis was done on split datasets: for signature assessment and mixture generation.

Comment 6 / Line 495: "cellular sub-types should only be included in deconvolution if they are > 2% abundant and < 95% correlated with other cell-types/sub-types": I find that this statement involves very arbitrary cut-offs as based on the results presented in supp fig 14 and 18: where the X-axis represents the mean abundance across the 100 simulated mixtures: how large is the variability of the abundance for each cell type. This is an oversimplified interpretation.

We thank the reviewers for the final suggestions, which we have addressed point-by-point below. The corresponding edits are highlighted in the Manuscript in red.

Reviewer #2 (Remarks to the Author):

I have last, but important comment related to a concern I raised during the first review. In my opinion, the ability to get accurate cell type-specific estimates from bulk data would be the strongest showcase of bulk deconvolution methods. Even through the authors added cell type-specific differential expression analysis of bulk data using CIBERSORTx in Supplementary Fig. 32, these important results are buried deep in the supplementary figures. These data should be added to one of the main figures.

As suggested, we have moved Supplementary Figure 32 in the main Results section as Figure 5.

Moreover, I think real, rather than simulated data should be used for this analysis. Authors could try generating pseudo-bulk estimates by pulling snRNA-seq profiles by sample and applying CIBERSORTx to pseudobulk. Additionally, CIBERSORTx could be applied to the autism and control bulk data from the Parikshak et al. This would greatly strengthen the case of applying bulk deconvolution methods in the era of single-cell genomics.

We have carried out the suggested pseudo-bulk analysis on ASD and control data from Velmeshev et al. and the results are now included in the Results section:

“Finally, we assessed cell-type specific DE in data from 15 Autism Spectrum Disorder (ASD) and 11 control samples from Velmeshev et al,⁴⁰, comparing results from cell-type-specific pseudo-bulked data versus that estimated by CIBERSORTx (Methods). Although no genes were significantly differentially expressed after multiple-testing correction, fold-changes determined in pseudo-bulk data correlated significantly with those estimated by CIBERSORTx for three of the four cell types (Methods; Supplementary Fig.32).

Overall, these results suggest that using cell-type specific gene expression for DE analyses is effective at detecting DE genes in the right cell-type when the gene expression and composition changes are not confounded, but this comes at the expense of a moderate increase in false-positives. Furthermore, low sample sizes and gene expression differences of low magnitude, as in the case of the pseudo-bulk ASD vs. control analysis, reduce the power of detecting cell-type specific DE genes.”

Given that the pseudo-bulk analysis suggested that cell-type specific analysis was not feasible in the case of weak fold-changes, we have not carried out further analysis of the Parikshak *et al.* data where the mean fold-change for composition-independent genes was 1.3.

Reviewer #3 (Remarks to the Author):

Comment 1: Some words on the complexity of the brains' transcriptome should also be given in the introduction, to explain why this manuscript focuses on brain tissue specifically.

As suggested, we have added the following paragraph in Introduction: “*For example, the human brain expresses the highest diversity of alternative splicing isoforms and non-coding RNAs²³, with single-cell sequencing now discovering hundreds of cell-types and cell-subtypes²⁴. We thus begin to address the question of tissue-specific properties of transcriptome deconvolution by focusing on the human brain.*”

Comment 3: Based on the new supplementary figure I do not agree that the mixtures of 100 cells from the Darmanis paper have a similar distribution compared to the bulk and other mixtures (lower log-cpm values).

We have rephrased the statement describing the data in Supplementary Figure 41, to reflect the reviewer’s comment: “*We confirmed that the CA- and VL-derived simulated single-nucleus mixtures had similar expression distributions to data from bulk brain tissue, whilst DM-derived simulated single-cell mixtures were not zero-inflated like single cells (Supplementary Fig.41).*”

Introduction (or discussion) misses part on previous benchmarking studies on computational deconvolution tools.

We have added the following statement in Introduction, to highlight the recent benchmarking studies, which we were already citing: “*Recent benchmarking studies have assessed the role of technical and biological factors in transcriptome deconvolution^{21,22}, but how these observations hold across tissues remains unclear.*”

Both in the main figure legends and the supplementary figure legends I sometimes miss annotation info (eg. what is CA, VL, TS, LK, NG, ...) and there are some inconsistencies in the use of abbreviations for example for Music (sometimes music and sometimes mus: main vs supplementary figures).

As suggested, we spelled-out the abbreviations in all figure legends, have limited the use of abbreviated labels where figure space permits, and ensured consistency across text and figures.

Line 158: I miss a more objective way to indicate which method performs best.

We have added the underlined text to specify the objective measures used to select the best performing method: “*CIBERSORT performing best ($r = 0.94$ and 0.95 for excitatory and inhibitory, respectively)*”

Line 161: why is the deconvolution performance comparison not performed using all methods (but only on 3 methods)? In this analysis, I also miss a mixture with low(er) abundant cell fraction, eg 10% vs 90% or 5% vs 95%...

The deconvolution on RNA mixtures was in fact performed using 6 of the 8 deconvolution methods: 3 partial deconvolution (Supplementary Fig. 7B), two enrichment methods (Supplementary Fig. 7C,D), and one complete deconvolution method (Supplementary Fig.28). *MuSiC* was not applied to the RNA mixture data because it required scRNA-seq, and *Coex* was not applied because it requires a higher number of samples than we have in the RNA mixture dataset.

The question of how lowly abundant cell types affect deconvolution accuracy is extensively addressed using *in silico* mixtures (Supplementary Fig.14, Supplementary.Fig18).

Line 163: “In both cases, the deconvolution accuracy was very high when the signature was derived from the same source as the mixtures”: what is meant by this sentence? Not clear to me. From a reply to one of my comments (comment 4), I thought that in this version analysis was done on split datasets: for signature assessment and mixture generation.

We apologize for the confusing statement. We meant that the signature and mixtures were derived from the same dataset, which was split. So the reviewer is right that the term “source” may be confusing here. We have rephrased to clarify: *“In both cases, the deconvolution accuracy was very high when the signature was derived from the same dataset as the mixtures”*

Comment 6 / Line 495: “cellular sub-types should only be included in deconvolution if they are > 2% abundant and < 95% correlated with other cell-types/sub-types”: I find that this statement involves very arbitrary cut-offs as based on the results presented in supp fig 14 and 18: where the X-axis represents the mean abundance across the 100 simulated mixtures: how large is the variability of the abundance for each cell type. This is an oversimplified interpretation.

While we agree that the stated cut-offs are somewhat arbitrary (like any cut-off), so we rephrased the text to allow readers to decide their own cut-offs: *“cellular sub-types should only be included in deconvolution if they are neither lowly abundant and nor highly correlated with other cell-types/sub-types”*